# The deubiquitinase Usp9x regulates PRC2-mediated chromatin reprogramming during mouse development

Trisha A. Macrae[1,2,3,4] & Miguel Ramalho-Santos [1,2,4,5✉]

Pluripotent cells of the mammalian embryo undergo extensive chromatin rewiring to prepare for lineage commitment after implantation. Repressive H3K27me3, deposited by Polycomb Repressive Complex 2 (PRC2), is reallocated from large blankets in pre-implantation embryos to mark promoters of developmental genes. The regulation of this global redistribution of H3K27me3 is poorly understood. Here we report a post-translational mechanism that destabilizes PRC2 to constrict H3K27me3 during lineage commitment. Using an auxin-inducible degron system, we show that the deubiquitinase Usp9x is required for mouse embryonic stem (ES) cell self-renewal. Usp9x-high ES cells have high PRC2 levels and bear a chromatin and transcriptional signature of the pre-implantation embryo, whereas Usp9x-low ES cells resemble the post-implantation, gastrulating epiblast. We show that Usp9x interacts with, deubiquitinates and stabilizes PRC2. Deletion of *Usp9x* in post-implantation embryos results in the derepression of genes that normally gain H3K27me3 after gastrulation, followed by the appearance of morphological abnormalities at E9.5, pointing to a recurrent link between Usp9x and PRC2 during development. Usp9x is a marker of "stemness" and is mutated in various neurological disorders and cancers. Our results unveil a Usp9x-PRC2 regulatory axis that is critical at peri-implantation and may be redeployed in other stem cell fate transitions and disease states.

[1] Eli and Edythe Broad Center of Regeneration Medicine and Stem Cell Research, University of California, San Francisco, San Francisco, CA, USA. [2] Center for Reproductive Sciences, University of California, San Francisco, San Francisco, CA, USA. [3] Medical Scientist Training Program, University of California, San Francisco, San Francisco, CA, USA. [4] Lunenfeld-Tanenbaum Research Institute, Mount Sinai Hospital, Toronto, ON, Canada. [5] Department of Molecular Genetics, University of Toronto, Toronto, ON, Canada. ✉email: mrsantos@lunenfeld.ca

mmediately after implantation, the pluripotent embryonic epiblast enters a period of accelerated growth. This amplification event corresponds to a transition in cell fate from a preimplantation state of naïve pluripotency to a postimplantation state of lineage induction. The stages of pluripotency can be modeled in vitro using mouse Embryonic Stem (ES) cells: culture in dual Mek/Gsk3β inhibition with leukemia inhibitory factor (2i/LIF) and vitamin C maintains a preimplantation-like state of pluripotency[1,2], while ES cells in serum/LIF mimic the fast-growing state of early postimplantation epiblast cells[3,4]. Primed ES cells in FGF2/Activin correspond to a later postimplantation stage, around mid-gastrulation[5].

Reprogramming of the chromatin landscape contributes to the transition in pluripotent cell states at peri-implantation[6]. This reprogramming event includes a global redistribution of the repressive histone mark H3K27me3, deposited by Polycomb Repressive Complex 2 (PRC2). Recent studies document that H3K27me3 forms broad genic and intergenic domains in pre-implantation embryos[7]. After implantation, H3K27me3 becomes concentrated over promoters of developmental regulatory genes[7,8], resembling patterns that restrain expression of bivalent (H3K27me3/H3K4me3-marked) genes in serum ES cells[9,10]. Similarly, H3K27me3 blankets the genome in 2i ES cells but is primarily promoter concentrated in serum[11,12]. The mechanisms that regulate this peri-implantation switch in PRC2 activity are unknown.

We recently reported a genome-wide screen that revealed that the chromatin state of ES cells is acutely tuned to variations in protein synthesis and degradation[4]. The deubiquitinating enzyme Ubiquitin Specific Protease 9x (Usp9x) was one of the top hits in this screen. Although its roles in chromatin regulation have not been investigated, Usp9x is a marker of "stemness"[13,14] and is a key, conserved regulator of several stem/progenitor cell types, including neural[15–17], hematopoietic[18], muscle[19] and intestinal cells[20]. For example, Usp9x promotes self-renewal of mouse neural stem/progenitor cells[15,21], and USP9X mutations are implicated in X-linked neurodevelopmental syndromes[22–24], Turner Syndrome[25], intellectual disability[26] and seizures[27] (reviewed in ref. [28].). Moreover, USP9X mutations occur frequently in human cancers[29].

We report here that Usp9x deubiquitinates and stabilizes PRC2, acting as a gatekeeper to the switch in H3K27me3 deposition patterns during mouse development. These findings shed light on the regulation of chromatin reprogramming in pluripotent cells and during lineage commitment, and have important implications for physiological and pathological settings where Usp9x and PRC2 are critical.

## Results

**Usp9x promotes ES cell self-renewal and a transcriptional state of preimplantation.** We sought a system to acutely deplete Usp9x in pluripotent cells. Usp9x-knockout ES cells minimally contribute to chimeric embryos[30] and demonstrate altered differentiation kinetics in vitro[31]. Because genetic deletions allow for cellular adaptation over time, we established an auxin-inducible degron (AID) system for acute control of Usp9x protein levels[32,33] (Fig. 1a). In ES cells homozygous for the OsTir1 auxin receptor, we tagged endogenous Usp9x with enhanced green fluorescent protein (GFP) and a minimal AID or 3x Flag tag (herein referred to as AID-Usp9x or Flag-Usp9x, respectively). Auxin drives substantial Usp9x protein depletion in AID-Usp9x cells within ~8 h (Supplementary Fig. 1a).

The AID tagging system enhances the underlying heterogeneity of Usp9x expression in ES cells cultured in serum/LIF, enabling us to use GFP expression to isolate subpopulations that resist degradation (Usp9x-high) or lose Usp9x (Usp9x-low) in response to auxin (Fig. 1a, Supplementary Fig. 1b), each fraction corresponding to ~20% of the total population. Usp9x-low ES cells display a 5-fold reduction in self-renewal capacity, measured by their ability to form undifferentiated, Alkaline Phosphatase (AP)-positive colonies when plated at low density (Fig. 1b). Knockdown of Usp9x by an alternative method (RNA interference) also induces loss of self-renewal (Supplementary Fig. 1c). Usp9x-high and Usp9x-low ES cells express comparable levels of Oct4, but Usp9x-low ES cells are Nanog-low (Fig. 1a), consistent with their self-renewal deficit. Interestingly, conditions that sustain ground state, naïve pluripotency (2i/LIF) impose homogeneity on Usp9x expression (Supplementary Fig. 1d). Usp9x has been found to interact with Oct4 and Sall4 as well as naïve factors Klf4 and Esrrb in mouse ES cells[31,34]. Taken together, these data suggest that Usp9x is coupled to the self-reinforcing naïve pluripotency network and may explain why a subset of AID-Usp9x ES cells in serum resist auxin-mediated degradation.

We hypothesized that the subset of cells in serum/LIF with high Usp9x expression and that resist its degradation are those that robustly express the naïve pluripotency network. We performed cell number-normalized (CNN) RNA-sequencing (RNA-seq) with spike-ins to characterize Usp9x-high and Usp9x-low ES cells. By principal component analysis (PCA), biological replicates cluster according to Usp9x levels (Fig. 1c). We calculated differential expression in 8 h Usp9x-high or Usp9x-low ES cells versus controls and compared their profiles to molecular signatures of development using Gene Set Enrichment Analysis (GSEA)[35] (Supplementary Data 1). This analysis revealed a striking polarity based on Usp9x levels: the Usp9x-high state correlates with preimplantation embryonic stages, whereas Usp9x-low ES cells resemble the postimplantation epiblast and early lineages (Fig. 1d). Usp9x-high ES cells express high levels of naïve state markers and low levels of primed state markers, while the opposite is observed in Usp9x-low ES cells (Supplementary Fig. 1e,f). Moreover, the expression of Usp9x itself declines from pre- to postimplantation in wild-type embryos (Fig. 1e and Supplementary Fig. 1g)[36,37] and with early lineage commitment by Embryoid Body (EB) formation (Supplementary Fig. 1h,i). Of note, low Usp9x expression does not represent a distinct cell cycle stage (Supplementary Fig. 1j). These results indicate that Usp9x promotes ES cell self-renewal and that loss of Usp9x captures the transcriptional reprogramming that occurs in pluripotent cells at implantation.

ES cells cultured in serum/LIF represent a heterogeneous mixture of interconvertible pluripotent states. Similar to the cases of naïve pluripotency markers such as Esrrb[38] and Nanog[39], isolated Usp9x-high or Usp9x-low cells do not readily redistribute along a spectrum of Usp9x expression after culture of the two populations in isolation. Rather, they adopt divergent phenotypes and undergo additional transcriptional reprogramming (Fig. 1f and Supplementary Fig. 2a,b). At 8 h, Usp9x-high ES cells demonstrate mildly decreased expression of many transcripts relative to no-auxin control cells, although only 277 down-regulated genes meet the threshold for differential expression (fold-change >1.5 and adjusted $P < 0.05$), with 70 upregulated genes (Supplementary Fig. 2c). By 48 h, Usp9x-high cells settle further into a state of hypotranscription[3], with suppression of the majority of the transcriptome relative to control cells (Fig. 1g and Supplementary Fig. 2c). Usp9x-low cells at 48 h no longer display a clear signature of primed pluripotency but show relative hypertranscription and induction of differentiation- and development-related Gene Ontology (GO) terms (Fig. 1g and Supplementary Fig. 1e, 2c-e). Downregulated genes are enriched for meiosis- and germline-related GO terms, reminiscent of the hypomethylated state of naïve pluripotency driven by vitamin C addition to 2i ES cell culture (Supplementary Fig. 2e). This

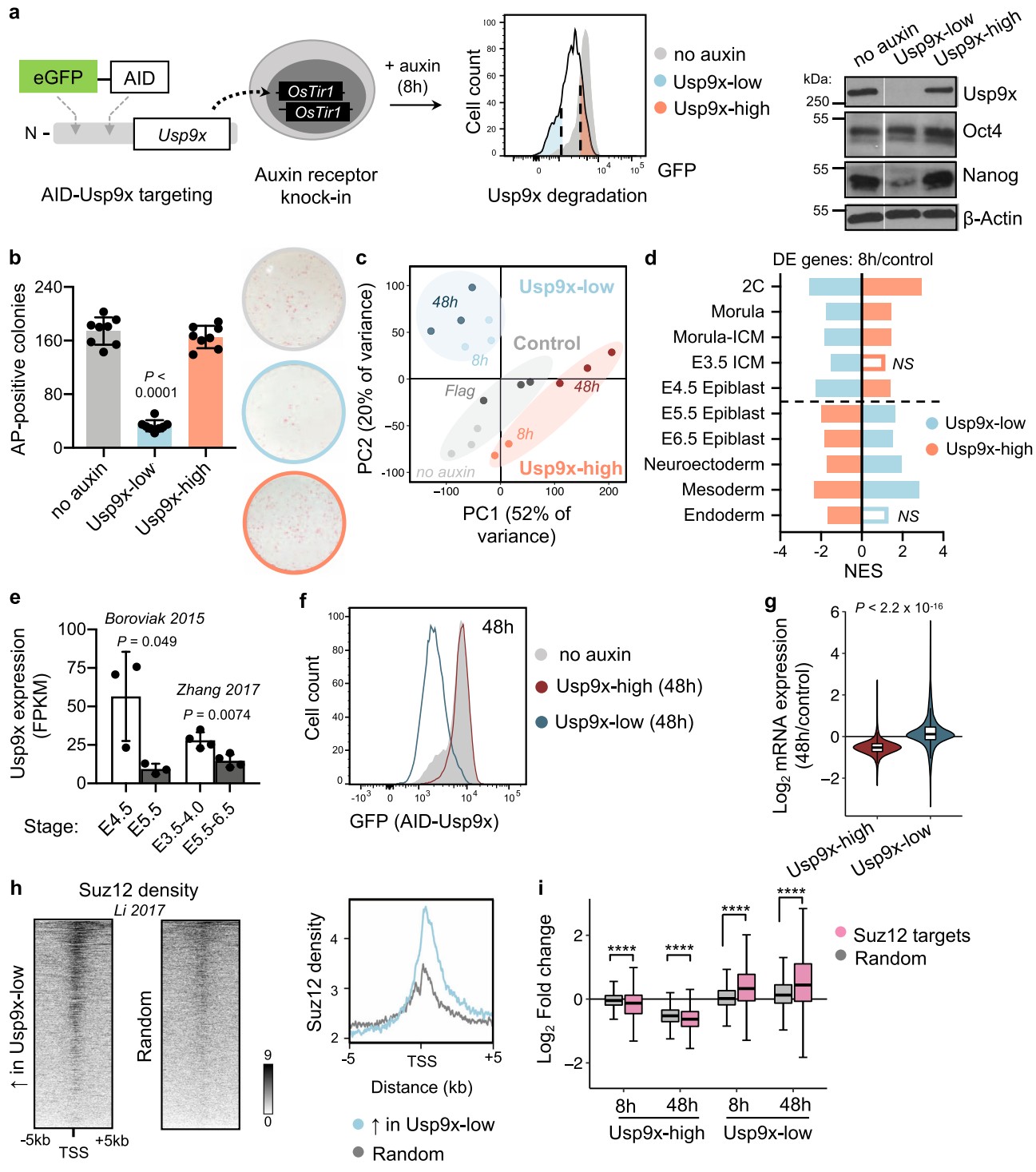

contrast parallels the predicted upregulation of transcriptional output along with lineage induction from the pre- to post-implantation epiblast[40,41]. These data further demonstrate that Usp9x levels mark distinct subpopulations of ES cells that diverge in their developmental potential.

We next probed the Usp9x-associated transcriptional signatures for clues to the regulatory networks establishing divergent cell fates. Consistent with their anticorrelated GSEA signatures (Fig. 1d), Usp9x-high and Usp9x-low ES cells show polarized expression of many of the same genes. Of the 277 genes significantly downregulated in Usp9x-high cells, 248 (90%) are significantly upregulated in Usp9x-low cells. ChIP-X Enrichment

Analysis (ChEA)[42] revealed that these genes are enriched for targets of Polycomb Repressive Complex 2 (PRC2) as well as Rnf2, a component of Polycomb Repressive Complex 1 (PRC1) (Supplementary Fig. 2f). In total, 65% (844/1310) of all the genes upregulated in Usp9x-low cells at 8 h are bound by Suz12 in ES cells (Supplementary Fig. 2f). Mapping Suz12 chromatin immunoprecipitation (ChIP)-seq signal[43] over these 1310 genes confirmed their enrichment for PRC2 binding (Fig. 1h), implicating PRC2 in the transcriptional polarity of Usp9x-high and Usp9x-low ES cells.

We asked whether Usp9x depletion relieves PRC2-mediated transcriptional repression. Suz12 target genes are upregulated

**Fig. 1 Usp9x promotes ES cell self-renewal and a transcriptional signature of preimplantation linked to PRC2 activity. a** Schematic of an auxin-inducible degron (AID) system for acute Usp9x depletion in mouse embryonic stem (ES) cells with representative flow cytometry plot of GFP (AID-Usp9x) expression in Usp9x-low and Usp9x-high ES cells. Right: western blot of endogenous Usp9x level in sorted cell fractions (see Supplementary Fig. 1b). **b** Quantification and representative images of colony formation assays. Usp9x-low ES cells display a self-renewal deficit. *AP*, Alkaline Phosphatase. **c** Principal Component (PC) Analysis of gene expression by RNA-seq. *8h*: 8 h auxin. *No auxin*: AID-Usp9x cells with vehicle treatment. *48h*: 8 h auxin followed by 48 h recovery without auxin. *Flag*: Flag-Usp9x cells after 8 h auxin and 48 h recovery. **d** The transcriptional signatures of Usp9x-high or Usp9x-low ES cells correlate with different stages of peri-implantation development by Gene Set Enrichment Analysis (GSEA). Genes differentially expressed between Usp9x-high or Usp9x-low ES cells and controls were used in each case. See Methods for references. DE, differentially expressed (relative to controls); NS, not significant (FDR > 0.05); NES, Normalized Enrichment Score. **e** Usp9x mRNA expression in the epiblast declines from pre- to postimplantation[36,37]. **f** Flow cytometry plot measuring median fluorescence intensity of GFP (Usp9x expression) in Usp9x-high and Usp9x-low ES cells after 8 h auxin treatment and 48 h recovery (without auxin). **g** Fold-change in expression of all genes at 48 h relative to control cells, showing hypotranscription in Usp9x-high ES cells and hypertranscription in Usp9x-low ES cells. **h** Heatmaps with summary profile plot of Suz12 binding (data from wild-type ES cells[43]) over the genes upregulated in Usp9x-low cells or a random subset ($N = 1310$). **i** Boxplots showing repression (in Usp9x-high) or induction (in Usp9x-low) of Suz12 target genes[57], compared to a random subset ($N = 3350$). Western blots represent at least two biological replicates (**a**). Data are mean ± s.d. of four replicates from two independent experiments (**b**), mean ± s.d. of 3–4 replicates (**e**), representative of three experiments (**f, h**). Boxplot hinges (**g, i**) show the first and third quartiles, whiskers show ±1.5*inter-quartile range (IQR) and center line shows median of three biological replicates. ****$P < 2.2 \times 10^{-16}$. *P*-values by one-way ANOVA with Dunnett's multiple comparisons test to the no-auxin condition (**b**), two-tailed Student's *t*-tests with Welch's correction (**e**), two-tailed Wilcoxon rank-sum test (**g**), and ANOVA with pairwise two-tailed Wilcoxon tests (**i**).

relative to a random gene subset in Usp9x-low ES cells, with reciprocal repression in the Usp9x-high state (Fig. 1i). Interestingly, genes bound by Suz12 alone versus Rnf2 and Suz12 (PRC1 and PRC2) show similar induction in Usp9x-low and repression in Usp9x-high ES cells, suggesting that co-binding of the complexes does not confer additional silencing (Supplementary Fig. 2g). Furthermore, expression of core PRC2 proteins declines with the transition from naïve (2i) to primed pluripotency (Supplementary Fig. 2h), along with Usp9x protein level and reported decreases in global H3K27me3[44,45]. Although PRC2-knockout ES cells can be propagated in 2i or serum conditions, deletion of core members (Suz12, Ezh2, or Eed) leads to a primed-like chromatin state[11] and induction of developmental regulatory genes[46,47]. In primed culture conditions, PRC2 deletions promote premature lineage commitment[48–50]. The fate of PRC2-knockout cells resembles that of Usp9x-low ES cells, which not only lose their ability to self-renew (Fig. 1b) but also display induction of primed pluripotency genes followed by spontaneous differentiation after auxin withdrawal (Supplementary Fig. 1e, 2b,e). Taken together, these data indicate that Usp9x-high ES cells represent a PRC2-repressed, preimplantation state of pluripotency, while activation of a subset of PRC2 targets in Usp9x-low ES cells promotes a postimplantation state of lineage induction.

**Usp9x-mutant embryos arrest at mid-gestation with incomplete repression of a subset of PRC2-targeted lineage genes.** We next turned to a mouse model to study the role of Usp9x in developmental progression. *Usp9x*-mutant embryos arrest at mid-gestation[30,51], although the stage and underlying causes of developmental arrest are unknown. To avoid confounding effects from roles of Usp9x in cleavage-stage and trophectoderm development[52,53], we used a Sox2-Cre to delete *Usp9x* strictly in the postimplantation epiblast derivatives of embryos[54]. We then genotyped and catalogued the morphology of control (ctrl, *Usp9x^fl/Y*) versus mutant (mut, *Usp9x^Δ/Y*) embryos at several mid-gestation stages (Fig. 2a, Supplementary Fig. 3a). Deviation from the expected (1:1) ratio arises by E11.5, at which point mutants account for only ~25% of living recovered embryos and have morphological abnormalities with 100% penetrance. The few mutants that survive to E11.5 show extensive hemorrhaging, while the others display pericardial edema, cerebral edema and severe delay, pointing to an earlier developmental arrest (Fig. 2b). *Usp9x* mutants already display developmental delay (delayed turning, open anterior neural tube) or gross abnormalities by

E9.5, including blunted posterior trunk development and exencephaly (Fig. 2b). These pleiotropic outcomes agree with the phenotypes of E9.5 chimeric embryos derived from *Usp9x*-gentrapped ES cells and the ubiquitous expression of Usp9x at E9.5[51,55].

*Usp9x* mutants appear morphologically normal at E8.5 (Supplementary Fig. 3b). We therefore performed whole-embryo RNA-seq at this stage to identify early transcriptional changes that may anticipate subsequent developmental defects. As expected, the transcriptional differences are relatively minor at this stage: we identified 71 upregulated and 66 downregulated genes across all *Usp9x* mutants (Fig. 2c, Supplementary Fig. 3c,d and Supplementary Data 2). Nevertheless, E8.5 *Usp9x* mutants are readily distinguished from controls by PCA and unsupervised hierarchical clustering (Supplementary Fig. 3e,f). Upregulated genes in *Usp9x* mutants are also upregulated in 48 h Usp9x-low ES cells (Supplementary Fig. 3g). These genes include targets of master developmental transcription factors (Isl1, Tfap2c, Eomes, Sox9 and Hnf4a, among others, Fig. 2d), and they are enriched for processes in cardiac/mesoderm and endoderm development (Supplementary Fig. 3h). Overall, the genes upregulated in *Usp9x* mutants at E8.5 typically decline by this point during wild-type development[56] (Fig. 2e), suggesting that Usp9x is required for appropriate silencing of developmental regulatory genes.

Incomplete repression of regulators of lineage commitment with defective differentiation is also observed in PRC2-hypomorphic ES cells[57,58]. We therefore probed the developmental dynamics of H3K27me3 levels at genes differentially expressed in *Usp9x* mutants. Recent ChIP-seq of wild-type mouse embryos documented a wave of H3K27me3 deposition during gastrulation[59]. Interestingly, the genes upregulated in E8.5 *Usp9x* mutants normally gain significant amounts of H3K27me3 between E6.5 and E8.5 (Fig. 2f), suggesting that PRC2 activity contributes to repressing them and/or reinforcing their transcriptional downregulation (Fig. 2e) at this stage.

Persistent expression of earlier developmental regulators may be a molecular harbinger of stalled development in *Usp9x*-mutant embryos, even though delay is not evident at the phenotypic level until E9.5. We speculate that incomplete repression of key genes, such as the TGFβ superfamily member *Nodal*, may impede developmental progression. *Nodal* normally accumulates H3K27me3 concurrent with its downregulation by E8.5 (Fig. 2g,h), but it remains upregulated in E8.5 *Usp9x* mutants (Fig. 2i). By contrast, the 66 genes downregulated in mutants are normally induced between E7.5 and E8.5 in wild-type development

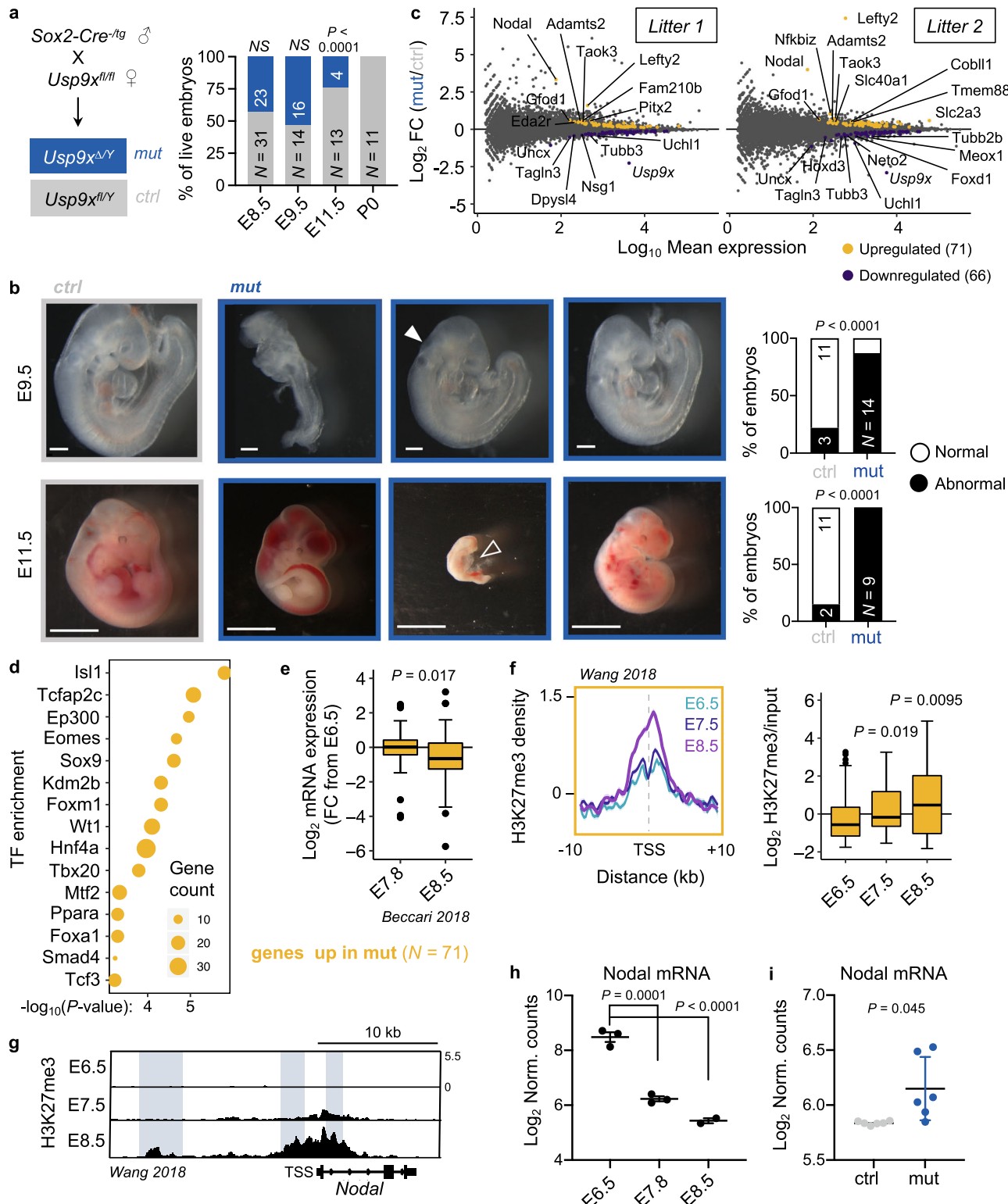

(Supplementary Fig. 3i-k), providing further transcriptional evidence of developmental delay prior to the emergence of phenotypic abnormalities. Taken together, these results indicate that E8.5 *Usp9x*-mutant embryos display persistent expression of early postgastrulation lineage commitment genes that normally gain H3K27me3 at this stage.

**Usp9x promotes a preimplantation pattern of H3K27me3 enrichment**. Our transcriptional analyses point to a role for

Usp9x in promoting PRC2-mediated silencing of developmental regulatory genes, both in ES cells to prevent their premature activation prior to lineage commitment (Fig. 1) and in postgastrulation embryos to allow for subsequent development (Fig. 2). We returned to ES cells to dissect the mechanism by which Usp9x regulates PRC2 activity. Consistent with their signature of PRC2 target gene derepression (Fig. 1i), Usp9x-low ES cells display globally reduced levels of H3K27me3 compared to Usp9x-high ES cells by cell number-normalized western blot

**Fig. 2 *Usp9x*-mutant embryos arrest at E9.5–E11.5 and display defective repression of early lineage programs marked by H3K27me3. a** Genetic cross to delete *Usp9x* in epiblast derivatives of postimplantation embryos. Quantification of recovered (live) male embryos at several postimplantation stages (right). **b** Sample images and quantification of control and mutant embryo phenotypes. Relative to controls (left), E9.5 embryos show variable developmental delay, with closed arrow indicating an open anterior neuropore. E11.5 embryos show a range of phenotypes, from hemorrhage to severe delay and death (tally includes dead embryos). Open arrow indicates pericardial edema. Scale bars = 250 µm (E9.5), 2.8 mm (E11.5), with *N* indicated. **c** MA plots of expression changes by RNA-seq in two litters of *Usp9x* mutants versus controls (at E8.5). 3 mutants and 3 controls were sequenced per litter (*N* = 12 embryos total; see Supplementary Fig. 3b-d). **d** Enrichr analysis of the top-enriched transcription factors (TF) that bind to the genes upregulated in *Usp9x*-mutant embryos in various cell types. **e** Expression of the 71 genes upregulated in *Usp9x* mutants during wild-type development[56]. *FC*, Fold-change relative to E6.5 embryos. **f** Distribution and boxplot quantification of H3K27me3 levels[59] over the promoters of genes upregulated in *Usp9x* mutants (10 kb upstream, 1 kb downstream of TSS). **g** Representative genome browser tracks of H3K27me3 in wild-type embryos (E6.5–E8.5) at the *Nodal* locus[59]. Known enhancer elements are highlighted and show gains of H3K27me3. **h** Nodal mRNA expression in wild-type development[56]. **i** Nodal mRNA expression in E8.5 *Usp9x*-mutant or control embryos. Boxplot hinges show the first and third quartiles, whiskers show ±1.5*IQR and center line shows median of 2–3 biological replicates (**e**, **f**). Data are representative of 2–3 biological replicates (**g**), mean ± s.e.m. of 2–3 biological replicates (**h**), or 6 biological replicates (**i**). *P*-values by $\chi^2$ test (**a**, **b**), Fisher's exact test (**d**), two-tailed Student's *t*-tests with Welch's correction (**e**, **i**), two-tailed Wilcoxon rank-sum tests (**f**), and ANOVA with Dunnett's multiple comparison test to E6.5 (**h**). $\chi^2 = 19.78$ (**a**), 85.19 (**b**, top), 147.8 (**b**, bottom).

(Fig. 3a). We next used spike-in normalized ChIP-seq to map H3K27me3 in Usp9x-associated cell states (Fig. 3b and Supplementary Fig. 4a). Compared to Usp9x-low ES cells, Usp9x-high ES cells display global gains of H3K27me3 at bivalent (dual H3K4me3/H3K27me3-marked) promoters (Supplementary Fig. 4b), which are canonical PRC2 targets highly enriched for developmental regulators. H3K27me3 gains are not limited to bivalent promoters, as Usp9x-high cells carry higher levels over peaks present at baseline (no-auxin condition) and spreading upstream and downstream (Fig. 3c,d and Supplementary Fig. 4b, c). A similar pattern is evident over and around PRC1 peaks (Supplementary Fig. 4d), consistent with biochemical evidence of the cooperation between PRC1 and PRC2[60]. Usp9x-high cells also have H3K27me3 enrichment over repetitive elements (Fig. 3e), which are targeted by this mark in naïve, preimplantation-like conditions[61]. Thus, the transition from Usp9x-high to Usp9x-low ES cells involves a genome-wide reduction in H3K27me3 across developmental genes and repeat elements, a shift from diffuse domains of H3K27me3 to peaks with minimal background.

The pattern of H3K27me3 in Usp9x-high serum ES cells resembles the diffuse domains of the mark in naïve (2i) ES cells and in preimplantation embryos[7,11,12]. In agreement with this notion, cumulative enrichment plots revealed that the global drop in H3K27me3 levels in Usp9x-high versus Usp9x-low ES cells recapitulates what is observed in the transition from 2i to serum ES cells and from preimplantation[62] to postimplantation embryos[59] (Fig. 3f). PCA of H3K27me3 ChIP-seq data separates preimplantation and postimplantation embryos along PC1. ES cell data also follow this trajectory. Usp9x-high and 2i ES cells align with preimplantation embryos, and Usp9x-low and serum ES cells cluster with postimplantation stages (Fig. 3g). Overall, no-auxin controls closely resemble Usp9x-low ES cells in their H3K27me3 distributions, suggesting that bulk sequencing of serum ES cells conceals the signal of broad H3K27me3 domains in the minority of cells with a naïve/preimplantation-like signature. These results indicate that H3K27me3 enrichment across large swaths of the genome (Fig. 3), together with a PRC2-repressed transcriptional program (Fig. 1), are hallmarks of preimplantation pluripotency[63,64] and Usp9x-high ES cells.

**Usp9x deubiquitinates and stabilizes PRC2.** We next explored the possibility that Usp9x may directly regulate PRC2 levels or activity. We found that the protein levels of core PRC2 components are downregulated in Usp9x-low ES cells (Fig. 4a, Supplementary Fig. 5a). This finding led us to hypothesize that Usp9x deubiquitinates and stabilizes PRC2 components to drive H3K27me3 deposition. In support of this notion, endogenous Usp9x interacts with Suz12 and Ezh2 in ES cell nuclear extracts

(Fig. 4b, Supplementary Fig. 5b). Moreover, acute AID-Usp9x depletion leads to the accumulation of poly-ubiquitinated forms of Suz12 and Ezh2 within 4–8 h of auxin addition (Fig. 4c). Alternative methods of reducing Usp9x activity, either small molecule inhibition (WP1130)[65] or overexpression of a mutant catalytic domain (C1566S), also lead to accumulation of ubiquitinated forms of Suz12 and Ezh2 (Fig. 4d and Supplementary Fig. 5c,d). These gains of ubiquitin upon Usp9x loss correlate with destabilization of Suz12 and/or Ezh2 (Fig. 4a,c,d and Supplementary Fig. 5c,d). Taken together, these data indicate that Usp9x interacts with PRC2 components and that its catalytic activity is required to promote a deubiquitinated state and higher protein levels of PRC2.

## Discussion

In summary, we report here that Usp9x deubiquitinates core PRC2 members to promote high levels of H3K27me3, repress developmental regulatory genes and maintain a preimplantation-like phenotype in ES cells (Fig. 5). Studies in mammalian systems have emphasized the developmental role of PRC2 in regulating bivalent promoters[9,10], which represent the highest-affinity sites for PRC2 activity[66]. However, H3K27me3 is widespread outside of bivalent chromatin[67]. Broad domains are evident over the genomes of preimplantation embryos and Usp9x-high ES cells and occur in other cell types later in development[68–71]. While the mechanisms underlying such promiscuity remain unclear, the Eed subunit has been shown to promote spreading of H3K27me3 domains[72]. PRC2 stability may also be a major factor[73]. For example, oncogenic *EZH2* mutations stabilize the complex and cause ectopic gains of H3K27me3 in lymphoma[74–76]. Promiscuous activity may be an ancestral function of PRC2[77], tuned and restrained by multiple layers of regulation. Our results highlight a mechanism whereby Usp9x stabilizes PRC2 to promote global increases in H3K27me3 and expansion to lower-affinity sites (Fig. 5).

It will be of interest to determine how the partnership between Usp9x and PRC2 integrates with the activity of other Usp9x substrates. Whereas prior studies indicate that *PRC2*-knockout ES cells remain pluripotent in serum[47], Usp9x-low ES cells derepress PRC2 target genes yet lose self-renewal capacity (Fig. 1). As a deubiquitinase with multiple cellular substrates[28,31], Usp9x may couple changes in the signaling milieu to the evolving chromatin landscape at peri-implantation, for example helping to modulate the transition from naïve to primed pluripotency networks in parallel with PRC2 activity. In addition, employing acute depletion models such as the AID may avoid genetic compensation and uncover novel aspects of Polycomb biology. It will also be important to identify the E3 ubiquitin ligase(s) acting on PRC2 in

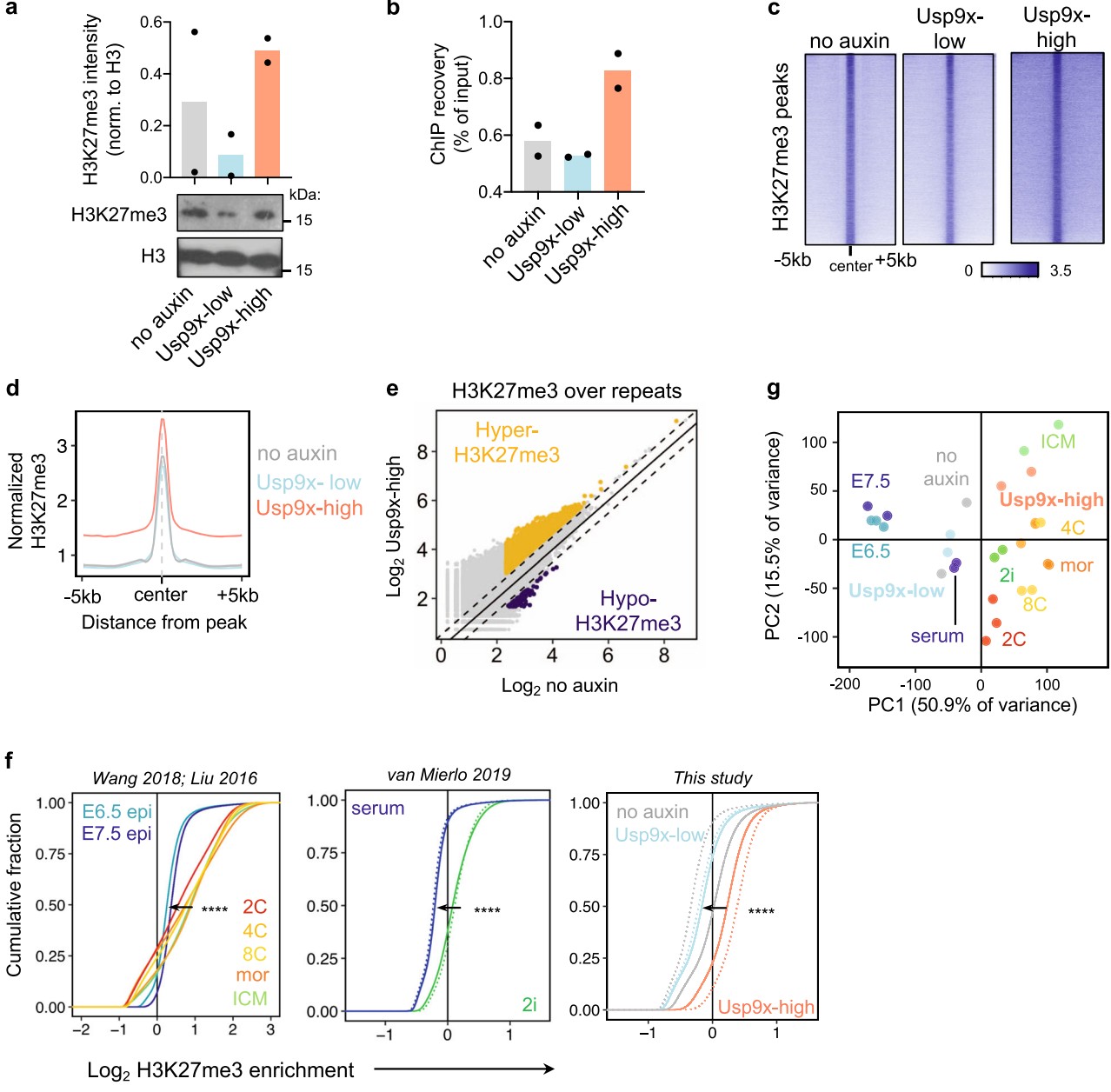

**Fig. 3 Usp9x mediates a pre- to postimplantation switch in global H3K27me3 levels. a** Quantification and representative western blot of H3K27me3 from histone extracts. **b** ChIP recovery before sequencing recapitulates the global gain of H3K27me3 in Usp9x-high ES cells. **c** Heatmaps of H3K27me3 ChIP-seq signal in Usp9x-high and Usp9x-low ES cells, showing H3K27me3 spreading in Usp9x-high cells. **d** Profile plot depicting the mean signal of coverage shown in **c**. **e** Usp9x-high ES cells carry more H3K27me3 over repetitive elements compared to untreated (no-auxin) cells. Each point represents an individual element. Dotted lines are $|\log_2(0.7)|$. **f** Cumulative enrichment plots of H3K27me3 enrichment in non-overlapping genomic bins of in vivo developmental stages (left) and ES cell states (middle, right)[11,59,62]. Preimplantation stages (2C-ICM, top) or preimplantation-like ES cell states (2i, Usp9x-high) display H3K27me3 enrichment. Epi, epiblast; mor. morula; ICM, inner cell mass. Enrichment (x-axis) is $\log_2$(H3K27me3/input $+0.5$). **g** PCA plot clustering Usp9x-high and Usp9x-low ES cells among the samples shown in **f** based on genome-wide H3K27me3 distributions. Each point represents a biological replicate. Data are mean of two biological replicates (**a, b**), sum of two biological replicates (**d, e**). ****$P < 2.2 \times 10^{-16}$ by two-tailed Kolmogorov–Smirnov test comparing the average of biological replicates.

ES cells. Several ligases have been found to ubiquitinate PRC2 in mammalian cells[78]. Our data reinforce that the balance between ubiquitin ligases and deubiquitinases dictates an important layer of PRC2 regulation during peri-implantation development.

What is the function of the broad domains of H3K27me3 in early development, if not to restrain bivalent gene expression? Maternally inherited H3K27me3 domains have been proposed to mediate noncanonical genomic imprinting in mouse embryos[79] and restrict enhancer expression in early-stage fly embryos[80]. Our

finding that Usp9x-high/PRC2-high ES cells enter a state of global hypotranscription (Fig. 1g,i and Supplementary Fig. 2c,d) raises the possibility that ubiquitous H3K27me3 in vivo suppresses large-scale transcription prior to implantation (Fig. 5), possibly by preventing H3K27 acetylation[81–83] and/or contributing to H2AK119ub deposition and chromatin compaction[84]. It will be of interest to investigate the interplay between H3K27me3 domains and PRC1 activity in the preimplantation embryo, especially in light of recent cell number-normalized data

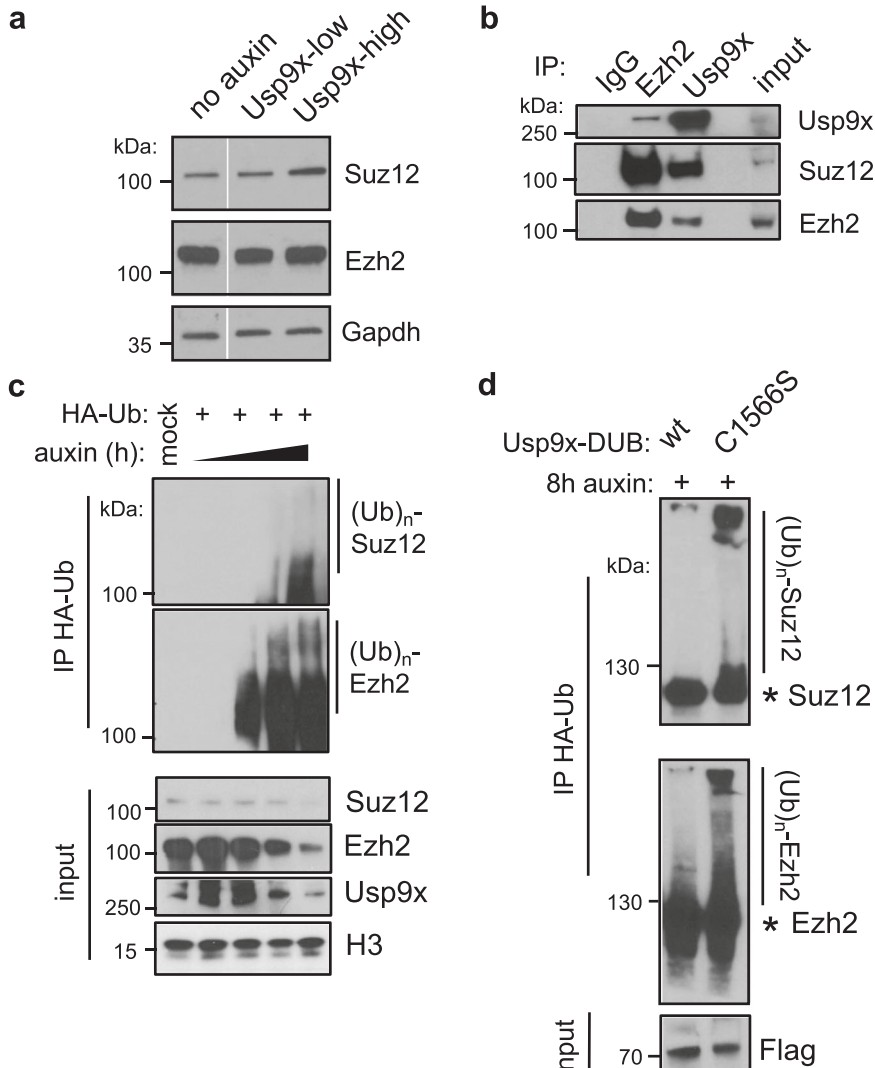

**Fig. 4 Usp9x is a PRC2 deubiquitinase. a** CNN western blots for Suz12 and Ezh2 proteins in whole-cell extracts. **b** Co-immunoprecipitation (*IP*) and western blot showing a reciprocal interaction between endogenous Ezh2 and Usp9x in wild-type ES cells. **c** Acute Usp9x depletion over a time course from 0–8 h leads to gain of ubiquitinated species of Suz12 and Ezh2 and destabilization of their protein levels. HA-Ub, HA-tagged ubiquitin; $(Ub)_n$, polyubiquitination. **d** Overexpressing a catalytic mutant (C1566S) versus wild-type (wt) form of the Usp9x catalytic domain (DUB) leads to gain of Suz12 and Ezh2 ubiquitin levels. AID-Usp9x cells were treated with auxin for 8 h to deplete endogenous Usp9x. Asterisk (*) designates the expected sizes for non-ubiquitinated species. Western blots are representative of at least two biological replicates.

documenting widespread H2AK119ub beyond Polycomb-enriched domains[85], similar to the diffuse patterns of H3K27me3 that we describe (Fig. 3). Extensive H3K27me3 may also safeguard embryonic potential during segregation of extra-embryonic lineages[63] or protect against uncontrolled activity of transposons during a period of low DNA methylation[61].

Our data suggest that the decline in Usp9x expression at implantation contributes to destabilizing PRC2 to allow exit from pluripotency and lineage induction. Usp9x is not the only factor that sets the threshold for PRC2 levels and activity at peri-implantation. We posit that post-translational modification augments other mechanisms that minimize H3K7me3 during this developmental window, including decreased transcription of Suz12;[86] rapid cell cycles that oppose H3K27me3 inheritance;[87,88] and accumulation of H3K27me3 antagonists, including RNA[41,89,90], activating chromatin marks[91,92] and DNA methylation[93,94]. Against this backdrop, PRC2 accessory proteins help recruit and retain the complex at CpG-rich regions such as bivalent promoters[43,67,94,95]. As a result, even Usp9x-low ES cells retain H3K27me3 over peaks—many of which are associated with promoters (Fig. 3c, d)—reflecting concentrated PRC2 activity at these sites.

The roles of PRC2 after gastrulation remain obscure. In zebrafish and Xenopus, H3K27me3 marks spatially regulated genes after gastrulation[96,97]. Intriguingly, recent work indicates that re-establishment of H3K27me3 after gastrulation may also contribute to spatial gene expression in mouse, including regulation of *Nodal*[98]. Constitutive *PRC2* deletions are peri-gastrulation lethal[99–101], but these findings are confounded by requirements for PRC2 in extraembryonic tissues[102], and the developmental consequences of epiblast-targeted *PRC2* knockouts are unknown. Together with the data presented here, these results suggest that Usp9x and PRC2 are redeployed to promote timely H3K27me3 deposition and silencing of key developmental genes after lineage commitment (Fig. 5). De novo H3K27me3 may also accumulate in the wake of decreased transcription to reinforce repression. Further studies are required to understand how Usp9x may regulate batteries of PRC2 target genes in a lineage-specific manner during organogenesis.

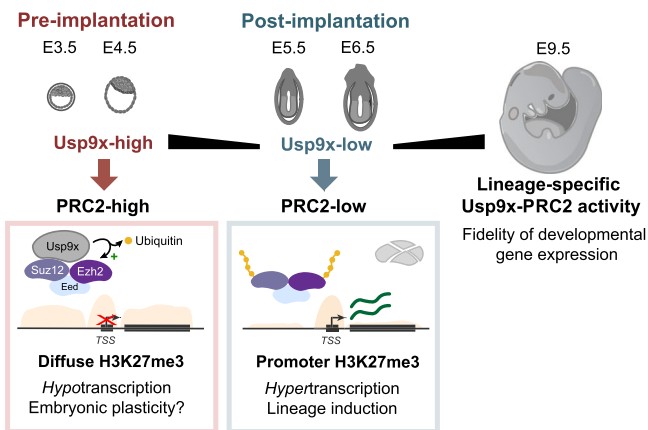

**Fig. 5 Model for the Usp9x-PRC2 regulatory axis in early development.** Usp9x is a PRC2 deubiquitinase that promotes diffuse H3K27me3 deposition and a preimplantation-like transcriptional state. Loss of Usp9x leads to PRC2 destabilization, restricted H3K27me3 deposition and global hypertranscription with priming of postimplantation lineages in ES cells. After gastrulation, *Usp9x* is required for timely silencing of developmental genes that are PRC2 targets and for normal development beyond E8.5, suggesting redeployment of the Usp9x-PRC2 connection later in development.

That *Usp9x* is X-linked prompts questions about its dosage during development. *USP9X* escapes X chromosome inactivation (XCI) in humans[103] and is a facultative escapee in mice[104]. Heterozygous loss-of-function *USP9X* mutations are implicated in sex-specific conditions, including a syndrome of developmental delay with congenital anomalies[23] and Turner syndrome[25]. Here, we focused on the outcomes of male (*Usp9x$^{-/Y}$*) mouse embryos because our in vitro studies were performed in male (XY) ES cells. Of note, *Usp9x* has a highly conserved homolog on the Y chromosome[105,106]. *Usp9y* is low-expressed in ES cells and bulk E8.5 embryos, and we did not detect its transcriptional upregulation after Usp9x depletion in either case, although we cannot exclude the possibility of compensation in other contexts. The proposed connection to PRC2, a regulator of XCI[79], raises the intriguing prospect that Usp9x may participate in mammalian dosage compensation.

The transition from a preimplantation to a postimplantation state of pluripotency mirrors stem cell expansion events in other compartments. Supporting the designation of Usp9x as a "stemness" factor[13,14], loss-of-function studies document that Usp9x restricts premature expansion of embryonic (this study), neural[15], hematopoietic[18], and intestinal stem/progenitor cell compartments[20] in mice. Similarly, PRC2 plays essential roles in each of these stem cell compartments[107–110]. In addition to neurodevelopmental conditions, mutations in *USP9X* are associated with several neurodegenerative disorders and cancers[22–24,26–29,111]. Thus, the Usp9x-PRC2 axis reported here merits further exploration in other developmental contexts and disease states.

## Methods

**Mice.** *Usp9x$^{fl/fl}$* females were maintained as homozygotes on a C57BL/6 background by crossing *Usp9x$^{fl/fl}$* and *Usp9x$^{fl/Y}$* mice[18]. Heterozygous male *Sox2-Cre* mice were obtained from Jackson Laboratories (JAX stock #008454) and bred with Cre-negative females to maintain a stock of heterozygous males[54]. All mice were housed at 65–75 °C with 40–60% humidity, maintained on a 12 h light/dark cycle and provided with food and water ad libitum in individually ventilated units (Techniplast) in specific pathogen-free facilities at The Center for Phenogenomics, Toronto. All procedures involving animals were performed according to the Animals for Research Act of Ontario and the Guidelines of the Canadian Council on Animal Care. The Animal Care Committee reviewed and approved all

procedures conducted on animals at TCP (Protocol 22-0331). Sample size choice was not predetermined.

Yolk sacs were dissected from embryos and used for DNA extraction with the Red Extract-N-Amp kit (Sigma). Usp9x status was assessed by PCR using Phire Green Hot Start II PCR Master Mix (Thermo Fisher Scientific). Cycling conditions: 98 °C for 30 s; 35 cycles of 98 °C for 5 s, 58 °C for 5 s, 72 °C for 8 s; 72 °C for 1 min. See Supplementary Data 3 for primer sequences.

**Plasmid construction.** An sgRNA was designed to target the *Usp9x* ATG with 20 nucleotide overhang in both directions. Cloning was performed by annealing pairs of oligos into pSpCas9(BB)-2A-GFP (PX458) (modified from GFP to BFP by site directed mutagenesis), a gift from Feng Zhang (Addgene plasmid # 48138; http://n2t.net/addgene:48138; RRID:Addgene_48138)[112]. Plasmid identity was verified by Sanger sequencing.

The eGFP-AID-Usp9x plasmid was assembled from pEN458-eGFP-AID[71-114]-eGFP, a vector carrying a minimal AID sequence for N-terminal targeting (a gift from Elphège Nora and Benoit Bruneau). The *Usp9x* N-terminus was chosen for targeting due to repetitive genomic sequence at the C-terminus. pEN458 was digested with NruI, NsiI and XhoI and gel purified to obtain the backbone and AID (Qiagen Gel Extraction kit). ~900 bp homology arms targeting the *Usp9x* ATG were amplified from mouse genomic DNA using PrimeStar GXL polymerase (Takara, CA, USA) and Gibson assembly primers with 21 nucleotide overlap to adjacent fragments. The purified vector fragments and homology arms were cloned by Gibson Assembly (NEB HiFi Assembly Kit).

Oligos containing a 3x FLAG sequence were annealed by incubation in standard annealing buffer (10 mM Tris-HCl pH 7.5, 1 mM EDTA, 50 mM NaCl) for 5 min at 95 °C followed by slow cooling to 25 °C. The annealed fragment was digested with BamHI and XhoI and purified by MinElute PCR purification (Qiagen). The eGFP-AID-Usp9x vector was also digested with BamHI/XhoI and cleaned up by gel extraction (Qiagen). The 3x Flag sequence was then ligated into the digested eGFP-Usp9x plasmid (Takara DNA ligation kit #6023).

**ES cell targeting.** Vectors were amplified by transformation into Stbl3 competent cells (Invitrogen). Resultant colonies were picked for miniprep DNA extraction (Qiagen) and screening by restriction enzyme digest. Positive clones were sequence-verified, purified by Maxiprep column extraction (Qiagen), concentrated by standard ethanol precipitation overnight, and used for nucleofection into low passage (p3) OsTir1-knockin ES cells derived as described[33]. Briefly, this line (148.4) was derived from E14 mouse ES cells and is homozygous for a Tir1-2A-Puro cassette (Addgene plasmid # 92142; http://n2t.net/addgene:92142; RRID: Addgene_92142) at the *TIGRE* locus.

Five million OsTir1 cells (passage 4) were nucleofected with 2.5 μg of the sgRNA plasmid and 20 μg of either eGFP-AID-Usp9x plasmid or eGFP-3x FLAG-Usp9x plasmid, using an Amaxa Nucleofector 2b device and ES nucleofection kit (Lonza) per the manufacturer's instructions. Cells were diluted in 500 μl of medium and immediately plated onto 10 cm dishes with 10 ml prewarmed medium. After 2 days, GFP-single and BFP-/GFP-double-positive cells were sorted by FACS and plated at clonal density (10,000 cells per 10 cm dish). Clones were left to expand for 5 days before manual picking onto 96-well plates. Single clones were then dissociated and expanded for 2 days. Clones were screened for auxin responsiveness by replica plating onto two 96-well plates, addition of auxin to 1 plate, and measurement of eGFP fluorescence intensity by flow cytometry. Auxin-responsive clones were subsequently expanded and used as biological replicates for all analyses. Cells were periodically pulsed with puromycin (1 μg/ml for 1–2 days) to select against transgene silencing.

**Usp9x-CD-mCherry cloning.** The Usp9x catalytic domain was amplified from a plasmid containing the full-length Usp9x ORF, obtained from DNASU;[113] mCherry was amplified by PCR from a pcDNA3-mCherry plasmid. We then cloned the purified Usp9x-DUB and mCherry fragments into pEF1a-IRES-Neo, a gift from Thomas Zwaka (Addgene plasmid # 28019; http://n2t.net/addgene:28019; RRID:Addgene_28019), by Gibson Assembly. To make the C1566S catalytic mutant form of the Usp9x-DUB domain, we performed site directed mutagenesis[114]. PCR was carried out with Phusion polymerase (New England Biolabs), with PCR cycling conditions: 98 °C for 7 min; 12 cycles of 98 °C for 30 s, 61 °C for 30 s, 72 °C for 3 min 45 s; 3 cycles of 98 °C for 30 s, 56 °C for 30 s, 72 °C for 3 m 45 s; 72 °C for 10 min; 4 °C hold. The PCR product was digested with DpnI for 3 h at 37 °C (New England Biolabs) and then 5 μl was transformed into Stbl3 competent cells (Thermo Fisher Scientific). See Supplementary Data 3 for primer sequences.

**Mouse embryonic stem cell culture.** ES cells were passaged every 1–2 days and grown in standard ES-FBS (serum/LIF) medium: DMEM GlutaMAX with Na Pyruvate, 15% fetal bovine serum (Atlanta Biologicals, GA, USA), 0.1 mM Non-essential amino acids, 50 U/ml Penicillin/Streptomycin, 0.1 mM EmbryoMax 2-Mercaptoethanol, and 1000 U/ml ESGRO supplement. For ground state culture conditions (2i), E14 ES cells initially grown in serum were passaged at least four times in 2i medium before use. 2i medium was composed of DMEM/F-12, Neurobasal, 1× N2/B27 supplements (Thermo Fisher Scientific), 1 μM PD0325901,

3 μM CHIR99021, 50 μM ascorbic acid, and 1000 U/ml ESGRO supplement. Cells tested negative for Mycoplasma contamination.

Indole-3-acetic acid sodium salt (Sigma I5148-2G) was dissolved in water to 500 mM, filter sterilized, and stored as aliquots at −20 °C. Stock solutions were thawed and diluted to 500 μM for all depletion experiments. Wild-type ES cells were used to determine the range of GFP-negative expression for sorting Usp9x-low ES cells upon auxin treatment. Usp9x-low and Usp9x-high cells correspond to the bottom and top ~15–20% of the population by GFP expression, respectively.

**Embryonic stem cell differentiation**. LIF was withdrawn from ES-FBS medium for spontaneous differentiation into Embryoid Bodies (EB). ES cells were counted, plated on low-attachment 6-well plates, and harvested at days 2 and 5 for analysis.

**siRNA-mediated knockdown**. siRNA transfections were performed in ES cells using Lipfectamine 2000 and OptiMEM (Thermo Fisher Scientific). ES cells were plated 5-7 h before transfection at a density of $5 \times 10^5$ cells per well of a 6-well plate and transfected with 100 pmol siRNA, according to the manufacturer's standard recommendations. A SMARTpool of four independent siRNAs was used to knockdown Usp9x (Dharmacon), and a nontargeting siRNA (siGenome siControl #2, Dharmacon) was used as a control. Transfections were performed in ES-FBS medium without antibiotics, and the medium was replaced the next morning with complete ES-FBS. Cells were harvested for counting and colony formation assays or western blots 48 h after transfection.

**Flow cytometry**. Cells were collected by trypsinization and resuspended in FACS buffer (10% FBS, PBS, with 500 μM auxin or water) and SYTOX Blue (Thermo Fisher Scientific) as a live-dead cell stain. Cells were sorted on the basis of GFP expression using the GFP-AID-Usp9x reporter. See Supplementary Fig. 6 for details. To initially define GFP-low/Usp9x-low populations, we used wild-type ES cells to determine the range of negative GFP expression. After 8 h of auxin treatment, an estimated 15–20% of auxin-treated cells fall into the GFP-low gate, with some biological variability between experiments. This range was subsequently applied to future experiments. The gate for GFP-high/Usp9x-high expression was set to collect a comparable fraction of the total population, 15–20%, at the upper end of GFP expression. The same number of intact GFP-Flag-Usp9x cells was sorted for RNA-seq. For analysis of Usp9x expression during the cell cycle, FUCCI ES cells[33] were sorted using mCherry as a marker of G0/G1 and BFP as a marker of S/G2/M. Sorts were performed on FACS AriaII (BD Biosciences) and Sony SH800 Single Cell Sorter (Sony) machines. Data were analyzed in FlowJo v10.4.2. All plots show median fluorescence intensity of the indicated marker.

**Colony formation assay**. One thousand cells from the indicated conditions were sorted and plated onto a 12-well plate. Four replicates were performed for two independent sorts. Cells were grown in self-renewal conditions (serum/LIF) for 5–6 days. Colonies were then washed 1× in PBS, fixed for 15 min at RT in 2% PFA, and stained according to the instructions of the VectorRed Alkaline Phosphatase (AP) Substrate Kit (Vector Laboratories, CA, USA) to stain undifferentiated colonies. Colonies were manually scored based on colony morphology and AP staining (positive if >50% of colony area).

**qRT-PCR**. cDNA synthesis was performed with the High-Capacity cDNA Reverse Transcription Kit (Thermo Fisher Scientific), using random hexamer priming for 2 h at 37 °C. KAPA 2x SYBR Green Master Mix, low ROX (KAPA) was used for qPCR and data were acquired on a QuantStudio 5 (Thermo Fisher Scientific) and analyzed in Prism v8 (GraphPad).

**RNA-seq library preparation**. Three independent clones of each cell line (AID-Usp9x or Flag-Usp9x) were used for RNA-sequencing. Cells were plated the day before sorting, and auxin was added to a final concentration of 500 μM in fresh media for 8 h, with water as a negative control. 250,000 live cells from each condition were sorted on the basis of negative SYTOX Blue incorporation, as above. For the 8 h timepoint, sorted cells were immediately pelleted, resuspended in Buffer RLT + β-mercaptoethanol (Qiagen), snap frozen on dry ice, and stored at −80 °C before library preparation. For the 48 h recovery timepoint, cells were re-plated in regular ES-FBS medium, cultured for 48 h without auxin, lysed, and snap frozen.

RNA extractions from frozen lysates were performed on the same day using RNeasy Mini columns (Qiagen). Recovered total RNA was quantified by Qubit and quality was assessed using an Agilent Bioanalyzer, RNA pico kit (Agilent). Synthetic RNAs from the External RNAs Control Consortium (ERCC) Spike-in Mix1 (Thermo Fisher Scientific) were added at known concentrations to the same volume of RNA from the previous step, per manufacturer's instructions (2 μl of 1:100 ERCC dilution added to 10 μl of RNA, equivalent to ~1–1.5 μg RNA). One microgram of total RNA was used for mRNA isolation and library preparation using the NEBNext Ultra II Directional Library Prep Kit for Illumina with the mRNA Magnetic Isolation Module, per manufacturer's instructions (New England Biolabs, NEB #E7420S and #E7490S). Library quality was assessed by Bioanalyzer High Sensitivity DNA chip (Agilent). Libraries were quantified by Qubit and pooled at equimolar concentration. Sequencing was performed on a HiSeq 4000

(Illumina) with 50 bp single-end reads at the UCSF Center for Advanced Technology.

For embryo RNA-seq, whole E8.5 embryos were dissected, cleaned of extraembryonic tissue, resuspended in buffer RLT + β-mercaptoethanol (Qiagen), and snap frozen on dry ice. Three litter-matched control and mutant embryos were collected from 2 litters, for a total of $n = 12$ individual embryos. RNA was extracted as above and 300 ng of total RNA was used for library preparation using the NEBNext Ultra II Directional Library Prep Kit for Illumina with the mRNA isolation module and NEBNext Multiplex Oligos for Illumina (New England Biolabs). DNA quality was assessed by Fragment Analyzer NGS (Agilent). Libraries were quantified by Qubit and pooled at equimolar concentration for sequencing on a NextSeq 500 (Illumina) with 75 bp single-end reads at the LTRI Sequencing Core.

**RNA-seq analysis**. Libraries were trimmed of Illumina adaptor sequences and quality-checked using Trim Galore! v0.4.0 (Babraham Bioinformatics), and then aligned to the mm10 transcriptome with ERCC sequences using TopHat2[115] v2.0.13 options -g 20 --no-coverage-search --library-type fr-firststrand --no-novel-indels. Gene counts were obtained from the featureCounts function of subread (v1.5.0) on the command line with options: -t exon -T 8 -s 2 -g gene_id. The table of raw counts was imported into R, filtered to remove low-count genes (genes with 0 counts in any sample and those with ≤3 counts per million, CPM by edgeR v3.26.8, across all samples were filtered out), and separated into ERCC and gene counts. Values for spike-in normalization were determined from ERCC counts corrected for overall library size using edgeR calcNormFactors: nf < −calcNorm-Factors(raw_ercc_counts, lib.size = N), where N < − colSums(raw_gene_counts). The CNN factors were then used to adjust gene counts using the limma-voom transformation[116] (option lib.size = N*nf). Data were further analyzed using tidyverse v1.2.1 and plotted using ggplot2 v3.3.2. The threshold for significant differential expression was adjusted $P < 0.05$ and log2FC > 0.7 or < −0.7 relative to control cells (AID-Usp9x without auxin and Flag-Usp9x). Boxplots and violin plots show fold-change relative to control cells obtained from toptable analyses.

For embryo RNA-seq, gene counts were obtained in the same manner, imported into R, and converted to a DESeq2 object (DESeqDataSetFromMatrix using sample information) for processing, DESeq2 version 1.24.0[117]. Genes with fewer than 10 raw counts across all samples were filtered out before differential expression analysis. Counts normalized by the DESeq2 rlog transformation were used for PCA and heatmaps of gene expression. Raw counts were used for differential expression analysis using the default parameters of the DESeq function. To account for staging differences between litters, we called differential expression between litter-matched mutants and controls, applied a statistical cutoff (adjusted $P < 0.1$), and overlapped the gene lists to obtain refined sets of up- or downregulated genes. RNA-seq data from stages E6.5, E7.8 and postoccipital E8.5 are from[56]. Published DESeq2 results were used to plot fold-changes in expression from E6.5, and raw fastq files were downloaded from NCBI GEO (GEO GSE113885), converted to normalized gene counts as above, and used to plot *Nodal* expression.

**Gene ontology and enrichment analyses**. Pathway analysis was performed by Gene Ontology (GO) analysis using DAVID 6.8 and geneontology.org[118–121]. P-values from GO analyses are based on Fisher's exact tests. Transcription factor binding enrichment of gene lists was performed with ChEA, part of the Enrichr suite (https://amp.pharm.mssm.edu/Enrichr/)[42,122]. Tables of enriched factors and P-values (determined from Fisher's exact tests) were downloaded. The top-enriched factors were selected based on adjusted P value, and enrichment was plotted in R using log₂ P values.

Gene Set Enrichment Analysis (GSEA v6.0.12) was performed using the online GSEAPreranked tool (https://cloud.genepattern.org/gp/pages/login.jsf)[35] with default conditions to compare differential expression (all genes sorted by log2FC) with gene sets from published datasets, outlined below. Normalized enrichment scores were plotted in Prism v8 (GraphPad).

*Datasets used for GSEA*. The 2-cell embryo signature is from[123]. Transcriptional signatures from cleavage stages through E5.5 were retrieved from[36], taking either the full gene list or the top 500 genes enriched for a particular timepoint from the published stage-specific expression analysis. The E6.5 epiblast signature was defined as genes upregulated in E6.5 epiblast relative to visceral endoderm and endoderm at E6.5[37]. A signature of early mesoderm was determined from published RNA-seq of ES cell differentiation[124], and we selected genes by fold-change of expression at the mesoderm stage compared to ES cells. The endoderm signature comes from published microarray data of early endoderm in E7.5 embryos[125]. Neuroectoderm genes were defined by RNA-seq data of epiblast stem cell differentiation to neural fate, comparing the fold-change in expression at day 2 of differentiation relative to baseline[126]. In all cases, either the full published gene list or the top 500 genes ranked by fold-change were used for GSEA.

**Suz12 ChIP-seq data analysis**. A normalized bigwig of Suz12 ChIP-seq in E14 ES cells, the combination of 3 biological replicates, is from[43] (GEO GSE97805). Coordinates for the 1310 genes upregulated in Usp9x-low ES cells (at 8 h) or a random subset of 1310 genes were converted to mm9 using the UCSC liftOver tool

(https://genome.ucsc.edu/cgi-bin/hgLiftOver), with the longest transcript selected to represent each gene. Suz12 coverage around TSSs was calculated using deeptools v3.3.0 computeMatrix (options reference-point -a 5000 -b 5000), and plots were produced using plotHeatmap and plotProfile.

For analysis of PRC2 versus PRC1 gene induction, lists of genes bound by Suz12 or Rnf2/Ring1b were obtained from[127]. We designated the overlap between these gene lists as Suz12/Rnf2-bound, in contrast to genes bound by Suz12 only or Rnf2 only. The $\log_2$ fold-change in expression for each of these genes, or all other genes expressed in ES cells, was determined from the results of toptable analysis and plotted in R.

**Histone extraction**. Histone extraction was performed using a standard acid extraction protocol[128]. Sorted cells were lysed for 10 min at 4 °C in triton extraction buffer (PBS with 0.5% Triton X-100, 2 mM PMSF, 1× Halt Protease Inhibitor at a density of $10^7$ cells/ml). Lysates were spun for 10 min at 4 °C, $1300 \times g$. The pellet was washed once in 0.5× volume of lysis buffer and centrifuged again. Pellets were resuspended in 0.2 N HCl ($10^6$ cells/ml) and acid extracted overnight, rotating at 4 °C. The next day, the solution was clarified by centrifugation and the supernatant transferred to a new tube. Histones were precipitated in 0.25× volume TCA, incubated 20 min on ice, and pelleted at max speed for 10 min. Excess acid was removed from solution through two washes in ice-cold acetone, pellets were air-dried, and histones were resuspended in water for BCA Protein quantification (Pierce). LDS sample buffer (Thermo Fisher Scientific) was added to 1× and samples were denatured for 5 min at 95 °C followed by cold shock.

**Co-immunoprecipitations**. Co-immunoprecipitation (Co-IP) assays were performed on nuclear extracts. ES cells grown to ~70% confluency were washed twice and then scraped in cold PBS. Cell pellets were weighed and resuspended in 4× volume of swelling buffer A (10 mM HEPES pH 7.9, 5 mM MgCl₂, 0.25 M Sucrose, 0.1% NP-40) with protease inhibitors added fresh (1× Halt Protease inhibitors, 1 mM PMSF, 1 mM NaF, 10 mM N-ethylmaleimide). Lysates were incubated on ice for 20 min and passed through a 18 ½ G needle five times. Nuclei were pelleted for 10 min at $1500 \times g$ and lysed in 8× volume buffer B (10 mM HEPES pH 7.9, 1 mM MgCl₂, 0.1 mM EDTA, 25% glycerol, 0.5% Triton X-100, 0.5 M NaCl with PIs as in buffer A). After incubation on ice for 10 min, samples were passed through an 18 ½ G needle five times and pulse sonicated using a probe sonicator, two times 5 s at 4 °C. One hundred microliter of lysate was diluted in 400 µl IP wash/dilution buffer (150 mM NaCl, 10 mM Tris pH 8, 0.5% sodium deoxycholate, 1% Triton X-100, 1 mM EDTA, 1 mM EGTA) and rotated 4 h to overnight with 1 µg Rb anti-Usp9x (Bethyl), 1.7 µg Rb anti-Ezh2 (CST #5246), or 1.7 µg Rb anti-IgG (Millipore CS200581). Input samples were collected at this time. Immune complexes were bound by 25 µl of prewashed Protein A Dynabeads (Thermo Fisher Scientific), rotating end-over-end for 2 h at 4 °C. Beads were washed in IP wash/dilution buffer, 3 × 5 min at 4 °C. Input and IP samples were eluted and denatured by boiling in 2× Laemmli buffer/bME for 10 min at 95 °C.

Co-IPs were also performed using Flag M2-bound magnetic agarose beads (Sigma) and GFP-Trap beads (ChromoTek). For Flag pull-downs, AID-Usp9x ES cells were used as controls for nonspecific binding to the Flag beads. For GFP pull-downs, the same amount of lysate was added to negative beads (ChromoTek) to control for nonspecific binding to beads. Cells were collected as above but diluted into GFP-Trap dilution buffer (10 mM Tris-HCl pH 7.5, 150 mM NaCl, 0.5 mM EDTA), immunoprecipitated by rotating for 1.5 h at 4 °C, and washed by 3× fast washes in GFP-Trap dilution buffer. Input and IP samples were denatured as above.

**HA-ubiquitin immunoprecipitations**. HA-tagged ubiquitin (a gift of the F. Sicheri lab) was overexpressed in ES cells by transfection with Lipofectamine 2000 (Thermo Fisher Scientific), 500 ng per $\sim 8 \times 10^6$ cells in a 10 cm dish. Water diluted in Lipofectamine was used for mock transfections. Medium was changed the next morning and cells were harvested after 24 h. Adherent cells were washed twice and then scraped into cold PBS. The resulting cell pellets were weighed and resuspended in 4× volume of RIPA buffer (150 mM NaCl, 1% NP-40, 0.5% Na deoxycholate, 0.1% SDS, 50 mM Tris pH 8) to lyse for 15 min on ice. Pellets were centrifuged at max speed for 10 min at 4 °C to remove insoluble material. One hundred microliter of supernatant was taken for IP and diluted to 500 µl in nondenaturing lysis buffer (20 mM Tris pH 8, 137 mM NaCl, 1% Triton X-100, 2 mM EDTA) plus 2.5 µg of anti-HA antibody (Abcam ab1190). IPs were incubated overnight at 4 °C with end-over-end rotation. The next day, immune complexes were bound to 25 µl Protein A Dynabeads (Thermo Fisher Scientific) for 2 h at 4 °C. Complexes were washed on beads for 3 × 10 min in IP wash buffer (150 mM NaCl, 10 mM Tris pH 8, 0.5% Na deoxycholate, 1% Triton X-100, 1 mM EDTA, 1 mM EGTA) and eluted in 2× Laemmli buffer/10% β-mercaptoethanol for 10 min at 95 °C followed by cold shock on ice. Input samples were collected and denatured in Laemmli buffer to 1×. Samples were removed from beads for western blotting.

For Usp9x catalytic domain expressions, transfections were performed as above but with the addition of 2.5 µg of plasmid (wild-type or C1566S pEF1a-Usp9x_CD-mCherry) and in medium without Pen/Strep. Transfection was checked by mCherry fluorescence the next morning. IPs were performed as above but with the following antibodies instead of HA: Ezh2 at 1:300 (CST #5246), Suz12 at 1:50 (CST #3737), or rabbit IgG at 1:50 (Millipore CS200581).

**Subcellular fractionation**. Subcellular fractionation was performed as previously reported[129]. Cell pellets were resuspended in buffer A (10 mM HEPES pH 7.9, 10 mM KCl, 1.5 mM MgCl₂, 0.34 M sucrose, 10% glycerol, 0.1% Triton X-100, 1 mM DTT, and PIs: NaF, PMSF, 1× Halt Protease inhibitor cocktail), incubated 5 min on ice, and centrifuged for 5 min at $1300 \times g$ at 4 °C. The supernatant was taken as the cytoplasmic extract and clarified by centrifugation. Nuclear pellets were washed in buffer A and resuspended in buffer B (3 mM EDTA, 0.2 mM EGTA, 1 mM DTT, and PIs). After 5 min on ice, chromatin pellets were centrifuged for 5 min at $1700 \times g$, 4 °C. The supernatant was collected as the soluble nucleoplasmic fraction. LDS sample buffer (Thermo Fisher Scientific) was added to 1× to cytoplasmic and nucleoplasmic fractions. Insoluble pellets were resuspended in 1× LDS sample buffer (Thermo Fisher Scientific) containing 5% β-mercaptoethanol and sonicated on a Bioruptor: high power, 30 s on, 30 s off, 5 min total (Diagenode). All samples were denatured for 5 min at 95 °C followed by cold shock.

**Western blot analysis**. Primary antibodies used in this study: rabbit anti-Usp9x, 1:2000 (Bethyl Laboratories, A301-351A); rabbit anti-H3K27me3, 1:1000 (CST #9733); rabbit anti-Suz12 1:1000 (clone D39F6, C ST #3737); rabbit anti-Ezh2, 1:2000 ((D2C9) XP CST #5246); rabbit anti-β-actin, 1:1000 (Abcam ab8227); rabbit anti-H3, 1:3000–1:5000 (Abcam ab1791); rabbit anti-Nanog, 1:1000 (CST #4903); rabbit anti-Oct4, 1:1000 (Santa Cruz SC-9081); mouse anti-Flag, 1:1000 (Sigma F1804); mouse anti-Gapdh, 1:2000 (Millipore MAB-374).

Western blots from sorted cells are all cell number normalized. Denatured samples were separated on 4-15% Mini-Protean TGX SDS-PAGE gels (Bio-rad). Protein was transferred to methanol-activated PVDF membranes (Bio-rad) by wet transfer (1x Pierce Transfer Buffer, 10% methanol) or using high molecular weight transfer conditions for the Bio-rad TransBlot Turbo (Bio-rad). Membranes were blocked in 5% milk/TBS-T and incubated with indicated primary antibodies for 1.5 h at room temperature or overnight at 4 °C. Membranes were then washed and incubated with HRP-conjugated anti-mouse/rabbit secondary antibodies, 1:10,000 (Jackson Labs) for 1 h at room temperature. Proteins were detected by ECL (Pierce) or Clarity (Bio-rad) detection reagents and exposure to X-ray film (Pierce). Quantification of western bands was performed using Fiji software[130]. Loading controls were performed on the same blot as the protein of interest except in cases where the proteins were similar sizes (e.g., β-actin and Nanog). In these cases, parallel blots were performed under the same conditions. Full scans of western blots as performed are provided with Source Data.

For cell cycle analysis, FUCCI reporter ES cells[33] were collected by trypsinization and sorted on a FACS AriaII (BD Biosciences) into mCherry + (G0/G1) and BFP + (S/G2/M) cell fractions. Cells were pelleted and lysed in RIPA buffer and clarified by centrifugation for 10 min, $13,000 \times g$ at 4 °C.

**H3K27me3 ChIP-seq**. Two biological replicates, consisting of independent clones of AID-Usp9x collected on consecutive days, were collected. $10^6$ cells were sorted and cross-linked in 1% formaldehyde/PBS, rotating for 10 min at room temperature. Cross-links were quenched with glycine (125 mM final) for 5 min at room temperature. Cells were washed 2× in cold PBS, snap frozen, and stored at −80 °C. All subsequent steps were performed on ice or at 4 °C. Fixed cell pellets were thawed and lysed in 1% SDS, 10 mM EDTA, 50 mM Tris-HCl pH 8 with protease inhibitors (1× Halt Protease inhibitor cocktail, 1 mM PMSF, 1 mM NaF) for 30 min. Chromatin was sheared to 200–500 bp fragments on a Covaris S220 with settings PIP 105, duty 2, cpb 200 for 9 min. Shearing efficiency was confirmed by 1% agarose gel electrophoresis. Chromatin lysates were clarified by centrifugation and diluted 1:10 in dilution buffer (1% Triton X-100, 2 mM EDTA, 167 mM NaCl, 20 mM Tris-HCl pH 8) with protease inhibitors. Inputs were collected at the same time. IPs were performed overnight using 2.5 µg of antibody (CST #9733 H3K27me3 or Abcam ab46540 rabbit IgG) per equivalent of 500,000 cells, rotating at 4 °C. The next day, immunocomplexes were precipitated by incubation with prewashed Protein A Dynabeads (Invitrogen) for 2 h. Beads were washed 4 × 10 min in low-salt buffer (0.1% SDS, 1% Triton X-100, 2 mM EDTA, 150 mM NaCl, 20 mM Tris-HCl pH 8), 1 × 10 min in high-salt buffer (0.1% SDS, 1% Triton X-100, 2 mM EDTA, 500 mM NaCl, 20 mM Tris-HCl pH 8), 1 × 10 min in LiCl buffer (0.25 M LiCl, 0.5% NP-40, 0.5% Na deoxycholate, 1 mM EDTA, 10 mM Tris-HCl pH 8), and 1× fast in TE. ChIP and input samples were eluted in fresh ChIP elution buffer (1% SDS, 50 mM NaHCO3, 50 mM Tris-HCl pH 8, 1 mM EDTA) and treated with RNase A for 1 h at 37 °C. Cross-links were reversed by shaking overnight at 65 °C with Proteinase K.

Genomic DNA was cleaned up using Qiagen MinElute Reaction Cleanup Kit (Qiagen) and quantified by Qubit (Thermo Fisher Scientific). ChIP efficiency was confirmed by H3K27me3 enrichment relative to IgG IP in qPCR at diagnostic regions. The same amount of chromatin from HEK293 cells (cell line source: ATCC) was spiked in to equivalent volumes of ChIP eluates (62 pg of spike-in chromatin per 25 µl of ChIP), yielding final concentrations between ~1–5%. Libraries were constructed from 2.5 ng of DNA and prepared using the NEBNext Ultra II DNA Library Prep Kit for Illumina with 9 PCR cycles (NEB #E7645S, New England Biolabs). Library quality was assessed by High Sensitivity DNA Assay on

an Agilent 2100 Bioanalyzer (Agilent Technologies). Samples were sequenced on a HiSeq 4000 using single-end 50 bp reads.

**H3K27me3 ChIP-seq data analysis**. Reads that passed quality control were trimmed of adaptors using Trim Galore! v0.4.0 and aligned to hg19 and mm10 using bowtie2 v2.2.5[131] with no multimapping. SAM files were converted to BAM files, sorted and indexed with samtools v1.9. Species-specific library sizes were determined from total mapped reads in hg19 bam files (using bowtie2 mapping statistics) and mm10 bam files (using samtools view -c). Normalization factors (NFs) are the ratio of mouse/human enrichment (mm10/hg19 reads) for each sample taken as a fraction of input (see Source Data), as in van Mierlo et al.[11]. ChIP-seq NFs correlate well with ChIP recovery as a fraction of input (Fig. 3b). Bam files were deduplicated for all downstream analyses, performed using picard v2.18.14 MarkDuplicates (http://broadinstitute.github.io/picard).

H3K27me3 ChIP-seq data from embryos were downloaded as fastq files from NCBI GEO. Preimplantation sequences are from Liu et al.[62] (GSE73952), postimplantation sequences from Wang et al.[59] (GSE97778), and wild-type ES cell data from van Mierlo et al.[11] (GSE101675). For paired-end samples, only one read was kept per fragment and all samples were trimmed, aligned to mm10, sorted and deduplicated as above. Deduplicated bam files were analyzed using deepTools v3.3.1 on the command line[132].

*Broad peak calling*. Deduplicated bam files were converted to scaled bedgraphs using deepTools v3.3.1 bamCoverage (options --scaleFactor <NF> --binSize 10 --blackListFileName ENCODE_mm10_blacklist.bed) and then to bed files: awk '{print $1"\t"$2"\t"$3"\tid-"NR"\t"$4"\t."}'. These scaled bed files were used to call broad peaks compared to input using epic2 on the command line (options -gn mm10, -d chrM). Bedtools (v2.28.0) merge was used to merge peaks within 3 kb, and bedtools intersect was used to determine a set of common peaks between replicates. Bam files were converted to scaled bigWigs using deepTools bamCoverage (options --binSize 100 --scaleFactor <NF>). Correlation between replicates was checked by multiBigWigSummary bins and plotCorrelation, and then scaled bw files were merged (bigwigCompare add) for heatmaps. computeMatrix was used to generate coverage of scaled bigwig files over no-auxin peaks (options scale-regions -m 500 --upstream 10000 --downstream 10000 --binSize 100 --missingDataAsZero --skipZeros --sortRegions descend --sortUsing mean --sortUsingSamples 1 -p max). Heatmaps were produced using deepTools plotHeatmap.

*Enrichment heatmaps and profile plots*. TSS profile plots were generated from the output of deepTools plotProfile (--outFileNameData), which was imported into R, processed to average replicates and plotted with ggplot2. For H3K27me3 coverage over *Nodal*, bigwig files were downloaded from Wang et al.[59] (GEO GSE97778). Sample tracks were visualized in Integrated Genome Viewer using bigwig files (IGV v2.3.92).

For PRC1 comparisons, BED files of Rnf2 and Cbx7 peak locations in E14 ES cells are from[133] (GEO GSE42466). BED files were converted to mm10 using UCSC liftOver. deeptools (v3.3.1) computeMatrix (options reference-point -a 5000 -b 5000 --referencePoint center --missingDataAsZero --skipZeros --smartLabels -p max) was used to compute normalized bigwig coverage of H3K27me3 over these peak regions. TSS profile plots were produced with plotProfile.

*Cumulative enrichment plots*. multiBamSummary was used to count H3K27me3 reads falling into non-overlapping 10 kb genes across the genome, and read counts were then imported into R. Embryo H3K27me3 ChIP-seq counts were normalized by library sizes (number of mapped reads in deduplicated bam files), and ES cell data were normalized by spike-in factors. For cumulative distribution plots, reads were counted in non-overlapping 10 kb genomic bins using deeptools multiBamSummary (options --smartLabels --blackListFileName --outRawCounts --minMappingQuality 10 -p max). The resulting counts table was imported into R, filtered to remove regions without coverage, scaled with the NFs calculated above and plotted using ggplot2 (stat_ecdf). P-values represent Kolmogorov–Smirnov test results using the averages of biological replicates. Counts per bin were adjusted for biological batch (embryo vs. ES cell origin) using ComBat/sva in R[134] and analyzed by PCA.

*Repetitive element analysis*. H3K27me3 was counted over repetitive elements annotated in the mouse genome (obtained from UCSC RepeatMasker) using featureCounts (options -f -O -s 0 -T 8). In R, we filtered out elements with low coverage, scaled using the NFs calculated above, and calculated the average of replicates. Plot shows regions with >5 normalized counts for hyper-H3K27me3 and <3 for hypo-H3K27me3 elements, with $|\log_2(\text{Usp9x-high/no-auxin})| >0.7$ as the threshold for enrichment.

*Boxplots of H3K27me3 enrichment*. For boxplot quantification of in vivo H3K27me3 signal over genomic regions, bedtools slop was used to define generous windows of promoter regions: TSS with 10 kb upstream and 1 kb downstream extensions (options -l 10000 -r 1000). deepTools multiBamSummary was used to calculate H3K27me3 from bam files over BED files of these promoter regions (options --minMappingQuality --outRawCounts). Raw counts for replicates were

imported into R, normalized to input and library size, averaged, and plotted as boxplots.

For analysis of H3K27me3 enrichment at bivalent genes, the H3K27me3 peaks that overlap known bivalent gene promoters in wild-type ES cells in serum were taken to be bivalent peaks. Specifically, bivalent promoters were defined from a list of bivalent genes[135] by bedtools slop (v2.28.0, options -l 1000 -r 100 -g < mm10_chrom_sizes> to extend TSS + 1 kb upstream and −100 bp downstream) and then intersected with H3K27me3 peaks defined in the no-auxin condition, as above (bedtools intersect with option -wb). deepTools multiBamSummary was used to count H3K27me3 reads falling in these 2566 bivalent peaks (options BED-file --smartLabels --outRawCounts -p 8) and over the list of all 17899 H3K27me3 peaks. Counts were downloaded for normalization and plotting in R.

**Reporting summary**. Further information on research design is available in the Nature Research Reporting Summary linked to this article.

## Data availability
RNA-seq and ChIP-seq data generated in this study have been deposited in Gene Expression Omnibus (GEO) under accession number GSE146800. Published datasets are available from Li et al.[43] (GSE97805), Beccari et al.[56] (GSE113885), van Mierlo et al.[11] (GSE101675), Wang et al.[59] (GSE97778), Liu et al.[62] (GSE73952), and Morey et al.[133] (GEO GSE42466). All relevant data supporting the key findings of this study are available within the article and its Supplementary Information files or from the corresponding author upon reasonable request. A reporting summary for this Article is available as a Supplementary Information file. Source data are provided with this paper.

## Code availability
Custom R scripts used for RNA-seq and ChIP-seq analyses are available on GitHub: https://github.com/tmacster/usp9x-project-scripts.

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

## Acknowledgements

We thank Aydan Bulut-Karslioglu and members of the Santos laboratory for input and critical reading of the manuscript. We are grateful to Elphège Nora and Benoit Bruneau for technical advice and for sharing the AID targeting vector and *OsTir1*-knockin ES cells. We also thank J. Lee and the Toronto Center for Phenogenomics for mouse colony support; M. Abed and V. Dixit for sharing the *Usp9x*-flox mice; E. Chow, K. Chan, and members of the UCSF Center for Advanced Technology and the LTRI Sequencing Core for next-generation sequencing; S. Biechele for cloning advice; and M. Percharde for bioinformatics guidance. Flow cytometry and sorting were performed in the UCSF Parnassus Flow Cytometry Core, supported by a Diabetes Research Center Grant and National Institutes of Health (NIH) grant P30 DK063720. This work

was supported by NIH grant 5F30HD093116 (to T.A.M.) and NIH grants R01GM113014 and R01GM123556, a Canada 150 Research Chair in Developmental Epigenetics and Project Grant 420231 from the Canadian Institutes of Health Research (to M.R.-S.).

## Author contributions

T.A.M. and M.R.-S. conceived of the project and wrote the manuscript. T.A.M. designed, performed and analyzed all experiments. M.R.-S. supervised the project.

## Competing interests

The authors declare no competing interests.
