## [Peer Review File · Nature Communications]

REVIEWER COMMENTS

Reviewer #1 (Remarks to the Author):

In this manuscript, Macrae and Ramalho-Santos find that the X-linked *Usp9x* gene is involved in deubiquitinating and thus stabilizing the developmental regulatory Polycomb Repressive Complex 2 (PRC2) in mouse embryonic stem cell (ESCs). Depletion of USP9X in mouse ESCs results in a transcriptional profile resembling primed pluripotent cells. Conversely, ESCs with higher levels of USP9X resemble naïve ESCs transcriptionally. The authors also show that when *Usp9x* is deleted in mouse embryonic epiblasts, development of the embryo is compromised. Consistent with a USP9X protein stabilizing the PRC2 histone methyltransferase enzyme EZH2, loss of USP9X resulted in decreased levels of EZH2 in ESCs. The authors further demonstrate that during post-implantation mouse embryonic development, a reduction in USP9X protein levels is associated with a reduction of the PRC2-catalyzed repressive histone H3K27me3 mark, thus leading to hypertranscription.

Overall, the strength of the manuscript is the discovery of a novel mechanism of the control of PRC2 function. Technically, the manuscript also generates a novel reversible USP9X protein depletion system by engineering an auxin-mediated protein degradation in ESCs. The work is an exciting addition to the PRC2 field. Below are comments that may help burnish the manuscript further.

Below are a set of comments on each of the main figures.

Figure 1: The authors used an auxin-inducible degron system to deplete USP9X in ESCs. When USP9X is depleted, the authors observed a decrease in expression of the pluripotency marker NANOG by Western blotting. Upon auxin induction, the ESCs can be partitioned into a high-USP9X and a low-USP9X populations of ESCs. USP9X-low cells have a self-renewal deficit and formed significantly less colonies compared to the USP9X-high colonies. Through RNA-seq, the authors identified that USP9X-high colonies were transcriptionally more similar to control ESC colonies, while USP9X-low colonies were transcriptionally more distinct and displayed gene expression signatures of primed pluripotent epiblast cells. The authors mined two publicly available datasets tracking gene expression during early development and found that normally *Usp9x* mRNA expression significantly decreases during the transition from a pre- to post-implantation state. Based on total transcription levels, they identified that USP9X-high cells were in a hypotranscriptive state and USP9X-low cells were in a hypertranscriptive state relative to controls. When comparing the differentially expressed genes between USP9X-high and USP9X-low cells, 248 genes were downregulated in the USP9X-high cells and upregulated in the USP9X-low cells. This set of genes were enriched for Polycomb targeted genes. These data indicate that a USP9X status correlates with a post-implantation expression profile and a state of hypertranscription, and that many of the differentially expressed genes in USP9X-high and -low cells are PRC2 targets.

Major Comments:

- a. The authors should justify why they are using an auxin inducible degron system, instead of generating *Usp9x*-null ESCs.
- b. The major concern with reduction/loss of USP9X protein is compromised pluripotency. There are many genes which are upregulated in UPS9X-low ESCs. Only a small subset of these genes is targeted by PRC2. The authors may want to include a discussion of why this is so and to what extent the dysregulated genes are a result of compromised pluripotency.
- c. The authors could also discuss why PRC2 mutant ESCs can self-renew but USP9X-low ESCs, on the other hand, appear to be compromised in self-renewal.
- d. The authors could also mine EZH2 protein expression and H3K27me3 levels in naïve vs. primed pluripotent cells to validate their model of *Usp9x* function.

Minor Comments:

- a. In Figure 1B the colonies are hard to see. Also, a better explanation of how the alkaline phosphatase assay measures self-renewal may be useful.
- b. Figure 1 legend should point out the color designations for USP9X-high, USP9X-low, and no

auxin used.

- c. If possible, the authors should quantify the western blot bands in Figure 1A.
- d. There should be an explanation of how cells were counted in Figure 1A either in the figure legend or text.

Figure 2: The authors generate a conditional *Usp9x* knock-out mouse model, where the *Usp9x* gene is only deleted in the post-implantation epiblast by a *Sox2-Cre* transgene. Mutant embryos are produced at a 1:1 ratio to controls around until embryonic day (E)11.5, when there are significantly fewer mutant embryos relative to controls. These mutant embryos displayed abnormalities such as hemorrhaging and developmental delay. The authors performed RNA-seq on the embryos and found that the mutant embryos were transcriptionally distinct from the control embryos already by E8.5. They performed an enrichment analysis on the upregulated genes in the mutant E8.5 embryos and found many developmentally regulated genes were upregulated. Using publicly available expression data, the authors found that these upregulated genes normally decrease in expression from E7.8 to E8.5. Based on another publicly available dataset, they identified that these upregulated genes normally progressively accumulate H3K27me3 from E6.5 to E8.5, and normally *Nodal* expression, which marks gastrulating embryos, decreases in expression from E6.5 to E8.5. However, in E8.5 embryos, *Nodal* expression is significantly higher in the mutant embryos compared to the controls. These data show that *Usp9x* is necessary for post-implantation development of the epiblast, and loss of *Usp9x* results in an upregulation of developmental regulator genes that normally decrease in expression and accumulate H3K27me3 during regular development.

Major Comments:

- a. The persistence of *Nodal* expression in the *Usp9x* mutant embryos may indicate a stalling of embryonic development. The authors may want to consider discussing this possibility.
- b. Perhaps beyond the scope of this study, but it would be interesting to know if there are differences between female and male embryos that lack *Usp9x*. The possibility that the Y-linked homolog of *Usp9x*, *Usp9y*, may compensate for the absence of *Usp9x* in males. Female embryos homozygous mutant for *Usp9x*, of course, will not be subject to such compensation.

Minor Comments:

- a. The number of samples should be included for the bar graphs in Figure 2B as they were in Figure 2A.
- b. Control and mutant embryos should be labeled in Figure 2B, because the colored borders don't stand out right away.
- c. It is unclear what the arrow is supposed to be pointing out in Figure 2B. This should be specified in the figure legend. Defects are also difficult to see in general and could be highlighted better.
- d. A list of genes dysregulated in the mutants may be useful, it is unclear what genes are dysregulated based on Figure 2C.

3. Figure 3: In *Usp9x*-low and -high ESCs, the authors they performed western blotting to show that global H3K7me3 is reduced in *Usp9x*-low cells compared to *Usp9x*-high cells. H3K27me3 is increased on repetitive elements in the *Usp9x*-low compared to *Usp9x*-high cells. By cumulative enrichment plot, the authors demonstrate that *Usp9x*-low cells had similar H3K27me3 enrichment to post-implantation E6.5 and E7.5 epiblasts, while *Usp9x*-high cells had similar H3K27me3 enrichment to pre-implantation states (2i, 2C, 4C, 8C, morula, E3.5 inner cell mass). The *Usp9x*-low cells clustered with these post-implantation states, while *Usp9x*-high cells clustered with pre-implantation states. These data show that *Usp9x*-low cells have a global reduction in H3K27me3, which correlates with a post-implantation state.

Minor Comment:

- a. If possible, quantification should be included for the western blot in Figure 3A.

4. Figure 4: The authors showed that PRC2 components *Suz12* and *Ezh2* were decreased in *Usp9x*-low cell by Western blot. They also showed that *Usp9x*, *Suz12* and *Ezh2* interact by Co-IP. They showed that as auxin is added to deplete *Usp9x* protein levels, ubiquitinated *Suz12* and *Ezh2* accumulate and *Ezh2* and *Usp9x* protein levels decrease. A similar result occurs when a *Usp9x*

catalytic mutant is expressed. These data suggest that Usp9x's deubiquitinase activity is required to prevent Suz12 and Ezh2 ubiquitination and a decrease in Suz12 and Ezh2 protein level.

Minor Comments:

- a. A "no auxin" control lane should be included for the western blot in Figure 4A. If possible, quantification should also be included for the band intensities in the western blot.
- b. The input lane has a very faint band for SUZ12 in Figure 4B. This should be repeated so the input bands are visible for all three proteins.

Supplement:

- a. In supplemental Figure 1G they should make it clear which genes are naïve or primed state markers.

Reviewer #2 (Remarks to the Author):

The authors clearly described the importance of USP9x in the maintenance of PRC2 expression levels and proper H3K27me3 deposition through preventing PRC2 from ubiquitin-mediated degradation. Also, they proved the biological significance of this process by revealing the loss of USP9x affected the self-renewal capacity of ESCs. The finding of this article is novel and interesting, and most of data demonstrated is decent, however, some of the data-interpretation and writing of the manuscript should be revised, and the mechanism that maintains certain amount of PRC2 and H3K27me3 in the Usp9x mutant or Usp9x-low cells should be clarified.

Major points

1. It seems a bit obscure why there are significant fractions of cells which harbors high levels of USP9x as authors showed substantial elimination of USP9x protein driven by the auxin-inducible degen system in Figure S1a. Why are the about 20% of the auxin-treated cells resistant to be degradation? Authors should discuss this issue in the texts. Also, if authors perform immunostaining with anti-Nanog antibody 8 hours after auxin treatment of the cells, do the cells indeed exhibit variation of the GFP signal and correlation with Nanog signal?
2. In line 49-52 the authors mentioned the re-distribution of the H3K27me3 during the transition in pluripotent cell states at peri-implantation which involves concentration of H3K27me3 over promoters of developmental regulatory genes, and it seemed as if they declared that they would tackle the mechanisms regulating this process. However, in Figure 3b and related texts they showed the weaker H3K27me3 signals in USP9x-lo cells, not the concentration to the promoters in USP9x-lo cells. Thus, the problems raised in the line 49-52 could be misleading.
3. Although authors focused on the relationship between Usp9x and PRC2, the component of PRC1, Rnf2 is also enriched in ChEA in Figure 1h. PRC1 and PRC2 are two major Polycomb repressive complexes and tightly linked together to repress developmental genes. Therefore, authors should investigate ChIP-seq of RNF2 and H2A mono-ubiquitination, a mark deposited by PRC1 after auxin-treatment as shown in Figure 3a-c to assess whether Usp9x is indeed connected to PRC2 alone or PRC1/PRC2.
4. In Figure 2g and 2h, Nodal already starts downregulation at E7.8 while H3K27me3 is not enriched yet at E7.5, suggesting that H3K27me3 itself does not play a role for the repression of Nodal. In line with this notion, I was wondering why H3K27me3 is maintained at high level around TSS in Usp9x mutant considering the finding that PRC2 could undergo strong poly-ubiquitination in the mutant. Authors should clarify why substantial amount of PRC2 can exist even in the Usp9x mutant and the mechanisms underlying thresholding of the level of PRC2. In my opinion, the distribution pattern of H3K27me3 is just the consequence rather than the cause of transcription. Authors should take into account those points and discuss in the text.

Minor points

1. Related to Major point 1. In line 52, although the authors attributed this re-distribution of H3K27me3 to switch in PRC2 activity, I think this could be overstated as grounds for this idea is not seems enough. Authors should modify texts.
2. I'm afraid enrich R is not yet common to all of us and thus Figure 1h right panel is not impressive. Rather, authors should show the distribution of PRC2 signals over promoters of DE (and other genes as negative controls) in the form of heatmaps.
3. I think the PCA analysis in Figure 2C is not so important. It'd be more convincing if authors showed the changes of expression of each 71 upregulated and 66 downregulated genes in the format of MA plot or so.
- 4, In the boxplots of Figure 2f right panel, the authors should state from where the signals are counted. Is that TSS±10 kb ?
- 5, In Figure 3e, it is unclear what the x-axis signifies.
6. In Figure 3f, authors should explain how they calculated the p-value.
7. In line 191 and 193, Nodal should be Nodal.
8. Authors should discuss what is the key player which promotes the ubiquitination of PRC2.

Reviewer #3 (Remarks to the Author):

Here, Macrae and Ramalho-Santos examine the functions of Usp9x in ES cell pluripotency and embryonic development. Using an auxin-inducible degron allele of Usp9x, they compare the effects of Usp9x loss to controls and Usp9x-high cells that are somehow insensitive to auxin addition. They show that Usp9x is required for ES cell self-renewal and promotes a transcriptional state representative of naïve ES cells and pre-implantation embryos, while Usp9x-low cells are more similar to post-implantation embryos. More generally, the Usp9x-high cells exhibit global hypotranscription, and the genes most affected by Usp9x levels are enriched for PRC2 binding. Consistent with these findings, Usp9x KO embryos are developmentally delayed and appear to incompletely repress early lineage commitment genes. The authors address the molecular mechanisms underlying these phenotypes and show Usp9x-high ES cells have higher overall H3K27me3 levels genome-wide, similar to pre-implantation embryos. Loss of Usp9x leads to accumulation of ubiquitinated forms of PRC2 subunits, suggesting Usp9x increases PRC2 levels and H3K27me3 by deubiquitinating PRC2 subunits.

Overall the manuscript is very thorough, using both ES cell and in vivo models to address the functions of Usp9x. The findings should be of significant interest to the stem cell and developmental biology communities. Although the experiments were largely well done and the data of high quality, I have a few questions and concerns, most notably regarding comparisons between Usp9x-high cells and other populations, as well as several minor points.

Major points:

1. Since many comparisons are made between control and Usp9x-high or -low cells, it is important to quantify Usp9x levels in each condition. However, the western blot in Fig. 1a, which suggests similar Usp9x levels between control and Usp9x-high cells, was run on two different gels. This prevents accurate comparison of the levels between control and Usp9x-high cells. Additionally, quantification of relative Usp9x levels in high, low, and no-auxin samples would be helpful.
2. Fig. 1g: If the Usp9x-high cells do indeed have levels of Usp9x that are similar to control cells, why do they exhibit hypotranscription relative to control cells? Unless I am missing something, this suggests the hypotranscription phenotype is independent of Usp9x levels. It is also strange that the hypotranscription of Usp9x-high cells relative to control cells is much stronger than the hypertranscription of Usp9x-low cells, which is very modest. Usp9x-low cells are strongly depleted

for Usp9x, but have relatively modest changes in mRNA levels, while Usp9x-high cells have a strong reduction in global mRNA levels despite expressing similar Usp9x levels as control cells. How can this be explained?

3. Similarly, in Fig. 3d, Usp9x-high cells exhibit higher H3K27me3 at repeat regions relative to no auxin controls. If Usp9x-high and no auxin controls have similar Usp9x levels, it is not clear what might be causing this increase in H3K27me3. Furthermore, it is odd that most of the comparisons in Fig. 3 are of Usp9x-high and Usp9x-low cells without the no-auxin control. Since the no-auxin control was profiled by ChIP-seq, it could be included in Fig. 3a-c, e-f. Because it is not clear why, on a molecular level, the Usp9x-high cells are resistant to auxin, comparisons with no auxin controls are important for the interpretation of these data.

4. In Fig3b-c, the local background (H3K27me3 levels outside of the peak regions) is higher in Usp9x-high cells to the same degree that the aggregate peak enrichment is higher. Could this be due to a normalization artifact, or do regions at least 5 kb from peaks actually exhibit increased H3K27me3 in Usp9x-high cells at many locations?

Minor points:

1. Fig. 1h: The number of genes observed and the number of genes expected from the 248 Usp9x affected that are targets of each regulator would be more informative than plots of their p-values, which have no biological meaning. Perhaps both sets of information could be included.

2. Fig. 2c: It is strange to show separate PCA plots for each litter. Presumably this is due to a batch effect that muddles the clustering, but plotting the data together would allow determination of exactly how well the two genotypes separate from each other in this analysis.

3. Quantification of the H3K27me3 levels in each population in Fig. 3a would help for understanding the magnitude of the global effect of Usp9x loss on this mark.

4. Fig. 4d: What is being immunoprecipitated and what antibody is being used in the western blot? The figure legend is not informative and the labels in the figure are lacking.

5. For some panels, the y-axis labels are confusing. For example, " $\log_2(\text{fold change}/\text{control})$ " suggests the fold change value is further divided by the control value.

We thank the reviewers for their helpful comments and suggestions. We have addressed each of the points raised by the reviewers, as detailed in the point-by-point response below. In summary, the major revisions are:

- (1) All three Reviewers prompted us to include no-auxin controls in relevant western blots and ChIP-seq analyses (Figures 3 and 4a) and this has now been done. Reviewers 2 and 3 also prompted us to better characterize why and how Usp9x-high ES cells differ from no-auxin controls despite having similar levels of Usp9x. We posit that the key point is not that no-auxin controls are the average of Usp9x *levels*, but rather the average of Usp9x-associated *cell states*. As bulk ES cells in serum, unsorted no-auxin controls are a metastable mixture of functionally heterogeneous cell states (e.g., reviewed in Hackett and Surani, *Cell Stem Cell*, 2014), including Usp9x-high and Usp9x-low subpopulations. We showed in the original submission transcriptional and phenotypic evidence that Usp9x-high ES cells resemble the ground state of naïve, pre-implantation pluripotency and Usp9x-low cells resemble a primed, post-implantation state (Figure 1d, Supplementary Figure 1e,f). We now reference a recent study that supports this notion, reporting that Usp9x interacts with naïve pluripotency factors Klf4 and Esrrb (de Dieuleveult et al., *bioRxiv*, 2020). Second, we show that culture conditions that promote a naïve, ground state of pluripotency (Mek/Erk and Gsk3b inhibition, 2i) impose unimodal Usp9x expression (new Supplementary Figure 1d), suggesting that Usp9x is downstream of the naïve pluripotency network. Per a suggestion from Reviewer 1 we use publicly available data to show that Usp9x protein level declines with the transition from naïve to primed pluripotency (new Supplementary Figure 2h). Together with the data from the initial submission, these data indicate that Usp9x is part of the self-reinforcing naïve pluripotency network, and its loss correlates with the exit from naïve pluripotency. Thus, a subset of heterogeneous cells in serum may be wired to resist its degradation.
- (2) In response to a comment from Reviewer 3, we performed additional analyses of hypo/hypertranscription in ES cells, included in the new Supplementary Figures 2c and 2d. Global shifts in steady-state transcript levels emerge between 8h and 48h, suggesting that important transcriptional reprogramming occurs during this time. At 8h, Usp9x level captures the transcriptional shift from naïve, pre-implantation-like pluripotency to primed, post-implantation-like pluripotency. By 48h, Usp9x-high ES cells display decreased expression of the majority of genes, more consistent with global hypotranscription than a specific transcriptional program. Usp9x-low ES cells show marked hypertranscription, with 2.5x more genes upregulated than downregulated (Supplementary Figure 2c) and with high expression of ribosomal protein genes (Supplementary Figure 2d). Together with the data from the original submission and the results outlined in point (1) above, these results indicate that global steady-state transcript levels do not directly correlate with Usp9x level but rather with Usp9x-associated cell states that have distinct developmental potential.
- (3) Reviewers 2 and 3 asked for additional clarity on the genes dysregulated in *Usp9x*-mutant embryos at E8.5. These data are enumerated in Supplementary Table 1 and aggregated in gene ontology/enrichment analyses, and, per suggestion of Reviewer 2, we now include MA plots in Figure 2 to highlight individual genes. In response to a question from Reviewer 1, we have also added additional text in the Results section for Figure 2 to mention the possibility that persistent expression of *Nodal*, one of the top genes upregulated across all mutants, may reflect developmental delay prior to the emergence of phenotypic abnormalities at E9.5.

(4) Reviewer 2 encouraged us to analyze the contribution of PRC1 to the proposed Usp9x-PRC2 axis, and we revisited our transcriptional and chromatin data and performed new analyses on this point. First, we found that targets of Suz12 alone versus Rnf2 *and* Suz12 (PRC1 *and* PRC2) are comparably repressed in Usp9x-high and derepressed in Usp9x-low ES cells (new Supplementary Figure 2g). Second, in Usp9x-high ES cells, H3K27me3 is enriched over PRC1 targets at similar levels to the rest of the genome (new Supplementary Figure 4d). These data lend further support to the notion that Usp9x promotes PRC2 activity at its typical sites of action in pre-implantation pluripotency, including the large fraction that are also targeted by PRC1.

We hope that the Reviewers agree that these revisions address their concerns and strengthen the manuscript, making it now suitable for publication in *Nature Communications*.

Reviewer #1 (Remarks to the Author):

In this manuscript, Macrae and Ramalho-Santos find that the X-linked *Usp9x* gene is involved in deubiquitinating and thus stabilizing the developmental regulatory Polycomb Repressive Complex 2 (PRC2) in mouse embryonic stem cell (ESCs). Depletion of USP9X in mouse ESCs results in a transcriptional profile resembling primed pluripotent cells. Conversely, ESCs with higher levels of USP9X resemble naïve ESCs transcriptionally. The authors also show that when *Usp9x* is deleted in mouse embryonic epiblasts, development of the embryo is compromised. Consistent with a USP9X protein stabilizing the PRC2 histone methyltransferase enzyme EZH2, loss of USP9X resulted in decreased levels of EZH2 in ESCs. The authors further demonstrate that during post-implantation mouse embryonic development, a reduction in USP9X protein levels is associated with a reduction of the PRC2-catalyzed repressive histone H3K27me3 mark, thus leading to hypertranscription.

Overall, the strength of the manuscript is the discovery of a novel mechanism of the control of PRC2 function. Technically, the manuscript also generates a novel reversible USP9X protein depletion system by engineering an auxin-mediated protein degradation in ESCs. The work is an exciting addition to the PRC2 field. Below are comments that may help burnish the manuscript further.

Below are a set of comments on each of the main figures.

Figure 1: The authors used an auxin-inducible degron system to deplete USP9X in ESCs. When USP9X is depleted, the authors observed a decrease in expression of the pluripotency marker NANOG by Western blotting. Upon auxin induction, the ESCs can be partitioned into a high-USP9X and a low-USP9X populations of ESCs. USP9X-low cells have a self-renewal deficit and formed significantly less colonies compared to the USP9X-high colonies. Through RNA-seq, the authors identified that USP9X-high colonies were transcriptionally more similar to control ESC colonies, while USP9X-low colonies were transcriptionally more distinct and displayed gene expression signatures of primed pluripotent epiblast cells. The authors mined two publicly available datasets tracking gene expression during early development and found that normally *Usp9x* mRNA expression significantly decreases during the transition from a pre- to post-implantation state. Based on total transcription levels, they identified that USP9X-high cells were in a hypotranscriptive state and USP9X-low cells were in a hypertranscriptive state relative to controls. When comparing the differentially expressed genes between USP9X-high and USP9X-low cells, 248 genes were downregulated in the USP9X-high cells and upregulated in the USP9X-low cells. This set of genes were enriched for Polycomb targeted genes. These data indicate that a USP9X status correlates with a post-implantation expression profile and a state of hypertranscription, and that many of the differentially expressed genes in USP9X-high and -low cells are PRC2 targets.

Major Comments:

a. The authors should justify why they are using an auxin inducible degron system, instead of generating *Usp9x*-null ESCs.

We thank the Reviewer for prompting us to clarify this point to the readers. Genetic knockouts enable study of full loss of gene function but run the risk of confounding from cellular adaptation and compensation. The auxin inducible degron (AID) system enables targeted depletion at the protein level, minimizing interference of transcription and translation. In the revised Results section for Figure 1, we now summarize these advantages as follows: “We sought a system to acutely deplete *Usp9x* in pluripotent cells. *Usp9x*-knockout ES cells minimally contribute to chimeric embryos (Nagai et al., *Mol Cell*, 2009) and demonstrate altered differentiation kinetics

in vitro (de Dieuleveult et al., *bioRxiv*, 2020). Because genetic deletions allow for cellular adaptation over time, we established an auxin inducible degron (AID) system for acute control of Usp9x protein levels (Fig. 1a).” Based on our experimental findings, we also note that “The AID tagging system enhances the underlying heterogeneity of Usp9x expression in serum ES cells, enabling us to use GFP expression to isolate subpopulations that resist degradation (Usp9x-high) or lose Usp9x (Usp9x-low) in response to auxin.”

b. The major concern with reduction/loss of USP9X protein is compromised pluripotency. There are many genes which are upregulated in UPS9X-low ESCs. Only a small subset of these genes is targeted by PRC2. The authors may want to include a discussion of why this is so and to what extent the dysregulated genes are a result of compromised pluripotency.

This is an important point of discussion. In fact, the majority of genes upregulated in Usp9x-low ES cells (~65%) are also bound by Suz12 in ES cells, highlighting the close relationship between Usp9x, PRC2, and pluripotency at the transcriptional level. We emphasize this point in Figure 1 and have added the following text to the Results section (new text underlined): “In total, 65% (844/1310) of all the genes upregulated in Usp9x-low cells at 8h are bound by Suz12 in ES cells (Supplementary Fig. 2f). Mapping Suz12 chromatin immunoprecipitation (ChIP)-seq signal over these genes confirmed their enrichment for PRC2 binding (Fig. 1h), implicating PRC2 in the transcriptional polarity of Usp9x-high and Usp9x-low ES cells.”

We also note that deletions of core PRC2 members induces primed-like chromatin and transcriptional programs in 2i or serum (van Mierlo et al., *Cell Stem Cell*, 2019; Leeb et al., *Genes Dev*, 2010; Chamberlain et al., *Stem Cells*, 2008) and premature differentiation in primed mouse ES cells (Shan et al., *Nat Comm*, 2017). This phenotype mirrors that of Usp9x-low ES cells, which demonstrate a primed-like transcriptional program at 8h and signs of early lineage commitment 48h after auxin withdrawal (Supplementary Fig. 1e, 2b).

c. The authors could also discuss why PRC2 mutant ESCs can self-renew but USP9X-low ESCs, on the other hand, appear to be compromised in self-renewal.

We discuss this point in greater detail in the Discussion (new text underlined): “It will be of interest to determine how the partnership between Usp9x and PRC2 integrates with the activity of other Usp9x substrates. Whereas prior studies indicate that PRC2-knockout ES cells remain pluripotent in serum, Usp9x-low ES cells derepress PRC2 target genes yet lose self-renewal capacity (Figure 1). As a deubiquitinase with multiple cellular substrates, Usp9x may couple changes in the signaling milieu to the evolving chromatin landscape at peri-implantation, for example helping to modulate the transition from naïve to primed pluripotency networks in parallel with PRC2 activity. In addition, employing acute depletion models such as the AID may avoid genetic compensation and uncover novel aspects of Polycomb biology.”

d. The authors could also mine EZH2 protein expression and H3K27me3 levels in naïve vs. primed pluripotent cells to validate their model of Usp9x function.

We now include quantitative proteomics data demonstrating decreased Ezh2/PRC2 protein expression from naïve to primed pluripotency (new Supplementary Fig. 2h). We also include references documenting a decrease in H3K27me3 levels during this developmental transition (Kurimoto et al., *Cell Stem Cell*, 2015; Tosolini et al., *Sci Reports*, 2018).

Minor Comments:

a. In Figure 1B the colonies are hard to see. Also, a better explanation of how the alkaline phosphatase assay measures self-renewal may be useful.

We have enlarged Fig. 1b and added an additional comment to the main text to reflect that AP stains undifferentiated colonies.

b. Figure 1 legend should point out the color designations for USP9X-high, USP9X-low, and no auxin used.

Figure 1 now includes additional clarity on color designations for the various cell populations.

c. If possible, the authors should quantify the western blot bands in Figure 1A.

We have added a subpanel to Supplementary Fig. 1b with quantification of replicate western blots.

d. There should be an explanation of how cells were counted in Figure 1A either in the figure legend or text.

The histogram in Fig. 1a is from flow cytometry of GFP expression. We have added an explanation in the figure legend.

Figure 2: The authors generate a conditional *Usp9x* knock-out mouse model, where the *Usp9x* gene is only deleted in the post-implantation epiblast by a *Sox2-Cre* transgene. Mutant embryos are produced at a 1:1 ratio to controls around until embryonic day (E)11.5, when there are significantly fewer mutant embryos relative to controls. These mutant embryos displayed abnormalities such as hemorrhaging and developmental delay. The authors performed RNA-seq on the embryos and found that the mutant embryos were transcriptionally distinct from the control embryos already by E8.5. They performed an enrichment analysis on the upregulated genes in the mutant E8.5 embryos and found many developmentally regulated genes were upregulated. Using publicly available expression data, the authors found that these upregulated genes normally decrease in expression from E7.8 to E8.5. Based on another publicly available dataset, they identified that these upregulated genes normally progressively accumulate H3K27me3 from E6.5 to E8.5, and normally *Nodal* expression, which marks gastrulating embryos, decreases in expression from E6.5 to E8.5. However, in E8.5 embryos, *Nodal* expression is significantly higher in the mutant embryos compared to the controls. These data show that *Usp9x* is necessary for post-implantation development of the epiblast, and loss of *Usp9x* results in an upregulation of developmental regulator genes that normally decrease in expression and accumulate H3K27me3 during regular development.

Major Comments:

a. The persistence of *Nodal* expression in the *Usp9x* mutant embryos may indicate a stalling of embryonic development. The authors may want to consider discussing this possibility.

This is an important consideration, given the well-described roles of *Nodal* in embryonic patterning and regulating the balance between mesendoderm vs neuroectoderm. We agree that ongoing expression of early developmental genes could be a consequence or a cause of developmental delay, with *Nodal* as a particularly promising candidate. We include additional discussion in the text for Figure 2 and Supplementary Figure 3 (new text underlined):

Incomplete repression of earlier developmental regulators may be a molecular harbinger of stalled development in *Usp9x*-mutant embryos, although delay is not evident at the phenotypic level until E9.5. We speculate that persistent expression of key genes such as *Nodal* may impede developmental progression. The TGF β superfamily member *Nodal* normally accumulates H3K27me3 concurrent with its downregulation by E8.5 (Fig. 2g,h), but it remains upregulated in E8.5 *Usp9x* mutants (Fig. 2i). The 66 genes downregulated in mutants are normally induced from E7.5-E8.5 (Supplementary Fig. 3i-k), providing further transcriptional evidence of developmental delay prior to the emergence of phenotypic abnormalities.

b. Perhaps beyond the scope of this study, but it would be interesting to know if there are differences between female and male embryos that lack *Usp9x*. The possibility that the Y-linked homolog of *Usp9x*, *Usp9y*, may compensate for the absence of *Usp9x* in males. Female embryos homozygous mutant for *Usp9x*, of course, will not be subject to such compensation. The reviewer raises an excellent point. We agree that this will be an interesting avenue for future study and have added a new paragraph to the Discussion on this topic. Specifically to the Reviewer's point we note: "For the current study, we focused on the outcomes of male (*Usp9x*^{-/-}) embryos because our *in vitro* studies were performed in male (XY) ES cells [...] *Usp9y* is low-expressed in ES cells and bulk E8.5 embryos, and we did not detect its transcriptional upregulation after *Usp9x* depletion in either case, although we cannot exclude the possibility of compensation in other contexts."

Minor Comments:

a. The number of samples should be included for the bar graphs in Figure 2B as they were in Figure 2A.

We have updated the graphs Fig. 2b as suggested.

b. Control and mutant embryos should be labeled in Figure 2B, because the colored borders don't stand out right away.

We have re-labeled Fig. 2b.

c. It is unclear what the arrow is supposed to be pointing out in Figure 2B. This should be specified in the figure legend. Defects are also difficult to see in general and could be highlighted better.

We have updated the labeling of Fig. 2b and enlarged the images to better demonstrate developmental defects.

d. A list of genes dysregulated in the mutants may be useful, it is unclear what genes are dysregulated based on Figure 2C.

In the new Fig. 2c, we now show the top dysregulated genes across all mutants in the form of MA plots. These plots provide further evidence that the degree of differential gene expression is relatively limited because we have captured the transcriptional landscape *before* the emergence of gross phenotype abnormalities in mutants, but several important developmental genes are dysregulated. Full lists of gene expression changes for both litters are provided in Supplementary Table 2.

3. Figure 3: In *Usp9x*-low and -high ESCs, the authors they performed western blotting to show that global H3K7me3 is reduced in *Usp9x*-low cells compared to *Usp9x*-high cells. H3K27me3 is increased on repetitive elements in the *Usp9x*-low compared to *Usp9x*-high cells. By cumulative enrichment plot, the authors demonstrate that *Usp9x*-low cells had similar H3K27me3 enrichment to post-implantation E6.5 and E7.5 epiblasts, while *Usp9x*-high cells had similar H3K27me3 enrichment to pre-implantation states (2i, 2C, 4C, 8C, morula, E3.5 inner cell mass). The *Usp9x*-low cells clustered with these post-implantation states, while *Usp9x*-high cells clustered with pre-implantation states. These data show that *Usp9x*-low cells have a global reduction in H3K27me3, which correlates with a post-implantation state.

Minor Comment:

a. If possible, quantification should be included for the western blot in Figure 3A.

We have performed quantifications of replicate western blots and include these data in Figure 3a.

4. Figure 4: The authors showed that PRC2 components Suz12 and Ezh2 were decreased in Usp9x-low cell by Western blot. They also showed that Usp9x, Suz12 and Ezh2 interact by Co-IP. They showed that as auxin is added to deplete Usp9x protein levels, ubiquitinated Suz12 and Ezh2 accumulate and Ezh2 and Usp9x protein levels decrease. A similar result occurs when a Usp9x catalytic mutant is expressed. These data suggest that Usp9x's deubiquitinase activity is required to prevent Suz12 and Ezh2 ubiquitination and a decrease in Suz12 and Ezh2 protein level.

Minor Comments:

a. A "no auxin" control lane should be included for the western blot in Figure 4A. If possible, quantification should also be included for the band intensities in the western blot.

We have updated Fig. 4a to include the no-auxin control lane and performed quantification of replicate blots, shown in Supplementary Figure 5a.

b. The input lane has a very faint band for SUZ12 in Figure 4B. This should be repeated so the input bands are visible for all three proteins.

A longer exposure of the IP has been substituted to demonstrate expression of Suz12 in the input lane.

Supplement:

a. In supplemental Figure 1G they should make it clear which genes are naïve or primed state markers.

Supplementary Figure 1g now indicates which genes are transcriptional markers of pre- or post-implantation.

Reviewer #2 (Remarks to the Author):

The authors clearly described the importance of USP9x in the maintenance of PRC2 expression levels and proper H3K27me3 deposition through preventing PRC2 from ubiquitin-mediated degradation. Also, they proved the biological significance of this process by revealing the loss of USP9x affected the self-renewal capacity of ESCs. The finding of this article is novel and interesting, and most of data demonstrated is decent, however, some of the data-interpretation and writing of the manuscript should be revised, and the mechanism that maintains certain amount of PRC2 and H3K27me3 in the Usp9x mutant or Usp9x-low cells should be clarified.

Major points

1. It seems a bit obscure why there are significant fractions of cells which harbors high levels of USP9x as authors showed substantial elimination of USP9x protein driven by the auxin-inducible degron system in Figure S1a. Why are the about 20% of the auxin-treated cells resistant to be degradation? Authors should discuss this issue in the texts. Also, if authors perform immuno-staining with anti-Nanog antibody 8 hours after auxin treatment of the cells, do the cells indeed exhibit variation of the GFP signal and correlation with Nanog signal?

Thank you for prompting us to better characterize Usp9x-high ES cells compared to their no-auxin counterparts. Our data indicate that Usp9x-high cells represent a distinct molecular state from Usp9x-low cells, resembling the pre-implantation, naïve state of pluripotency. We show that Usp9x correlates with Nanog expression in sorted cell fractions (Fig. 1a), and to the

Reviewer's point we now include additional evidence that helps situate Usp9x in the naïve pluripotency network, beyond a strict correlation to Nanog levels. Additionally, we mined publicly available data to show that Usp9x protein expression declines during the transition from naïve to primed pluripotency, shown in the new Supplementary Fig. 2h. This may contribute to their distinct response to auxin-mediated degradation. We include new data and discussion in the Results section (page 3) as follows:

“A major advantage to the AID system compared to a genetic knockout is that it captures the underlying heterogeneity of Usp9x expression in serum ES cells. [...] Usp9x-high and Usp9x-low ES cells express comparable levels of Oct4, but Usp9x-low ES cells are Nanog-low (Fig. 1a), consistent with their self-renewal deficit. Interestingly, conditions that sustain ground state, naïve pluripotency (2i/LIF) also impose homogeneity on Usp9x expression (Supplementary Fig. 1d). Usp9x has been found to interact with Oct4 and Sall4 as well as naïve factors Klf4 and Esrrb in mouse ES cells (van den Berg et al., *Cell Stem Cell*, 2010; de Dieuleveult et al., *bioRxiv*, 2020). Taken together, these data suggest that Usp9x is coupled to the self-reinforcing naïve pluripotency network and may explain why a subset of AID-Usp9x cells in serum/LIF resist auxin-mediated degradation.”

2. In line 49-52 the authors mentioned the re-distribution of the H3K27me3 during the transition in pluripotent cell states at peri-implantation which involves concentration of H3K27me3 over promoters of developmental regulatory genes, and it seemed as if they declared that they would tackle the mechanisms regulating this process. However, in Figure 3b and related texts they showed the weaker H3K27me3 signals in USP9x-lo cells, not the concentration to the promoters in USP9x-lo cells. Thus, the problems raised in the line 49-52 could be misleading.

We note that studies of H3K27me3 in embryos have so far precluded the use of cell number normalization and/or spike-ins that would be important for careful quantitative comparisons of pre-implantation versus post-implantation patterns of the mark. The best quantitative model that we have for this transition is ES cells (van Mierlo et al, *Cell Stem Cell*, 2019; Kumar et al., *Cell Reports*, 2019), where H3K27me3 remodeling from 2i to serum ES cells closely resembles the patterns we describe in Usp9x-associated cell states, even though Usp9x-high cells are still grown in serum conditions. To address the reviewer's concern, we have modified the text in several locations (new text is underlined):

- (1) In the Introduction: “Recent studies document that H3K27me3 forms broad genic and intergenic domains in pre-implantation embryos. After implantation, H3K27me3 becomes concentrated over promoters of developmental regulatory genes, resembling patterns that restrain expression of bivalent (H3K27me3/H3K4me3-marked) genes in serum ES cells. Similarly, H3K27me3 blankets the genome of 2i ES cells but is primarily promoter-concentrated in serum.”
- (2) In the Results section: “Thus, the transition from Usp9x-high to Usp9x-low ES cells involves a genome-wide reduction in H3K27me3 across developmental genes and repeat elements, a shift from diffuse domains of H3K27me3 to peaks with minimal background.”
- (3) Additionally, we have revised the heading of the Results section pertaining to Figure 3 to: “Usp9x promotes a pre-implantation pattern of H3K27me3 enrichment”; and updated the figure legend to emphasize the changes to the global pattern of genome-wide enrichment of H3K27me3.

3. Although authors focused on the relationship between Usp9x and PRC2, the component of PRC1, Rnf2 is also enriched in ChEA in Figure 1h. PRC1 and PRC2 are two major Polycomb repressive complexes and tightly linked together to repress developmental genes. Therefore,

authors should investigate ChIP-seq of RNF2 and H2A mono-ubiquitination, a mark deposited by PRC1 after auxin-treatment as shown in Figure 3a-c to assess whether Usp9x is indeed connected to PRC2 alone or PRC1/PRC2.

We thank the reviewer for prompting us to look into the relationship between Usp9x and PRC1. We agree that an in-depth investigation of the interplay between Usp9x, PRC2, and PRC1 would be an interesting avenue for future investigation. To address this question in a more direct way, we first asked if our transcriptional data could provide a functional readout of the connection between Usp9x and PRC1. In particular, we asked if Suz12/Rnf2-bound genes show greater transcriptional change than Suz12-only targets, which would point to a cooperative relationship between the two complexes. This analysis is included in the new Supplementary Fig. 2g, where we report that “genes bound by Suz12 alone versus Suz12 (PRC2) and Rnf2 (PRC1) show similar induction in Usp9x-low and repression in Usp9x-high ES cells, suggesting that co-binding of the complexes does not confer additional silencing (Supplementary Fig. 2g).”

In addition, we mined our existing H3K27me3 ChIP-seq data for evidence that the distribution of the mark is responsive to PRC1 activity. As the reviewer notes, the activities of PRC1 and PRC2 are closely linked in mouse ES cells, and they share numerous targets. It is expected that high levels of H3K27me3 would recruit PRC1. We compared H3K27me3 ChIP-seq signal over Cbx7 (canonical) versus Rnf2 (all PRC1), shown in the new Supplementary Fig. 4d. Just as Usp9x-high ES cells have high levels of H3K27me3 over, upstream, and downstream of H3K27me3 peaks, we now note that “A similar pattern is evident over and around PRC1 peaks (Supplementary Fig. 4d), consistent with biochemical evidence of the cooperation between PRC1 and PRC2.”

Finally, we revised the Discussion section to include the following (new text underlined): “Our finding that Usp9x-high/PRC2-high ES cells enter a state of global hypotranscription (Fig. 1g,i and Supplementary Fig. 2c,d) raises the possibility that ubiquitous H3K27me3 *in vivo* suppresses large-scale transcription prior to implantation (Fig. 5), possibly by preventing H3K27 acetylation and/or contributing to H2AK119ub deposition and chromatin compaction. It will be of interest to investigate the interplay between H3K27me3 domains and PRC1 activity in the pre-implantation embryo, especially in light of recent cell number-normalized data documenting widespread H2AK119ub beyond Polycomb-enriched domains, similar to the diffuse patterns of H3K27me3 that we describe (Fig. 3).”

4. In Figure 2g and 2h, Nodal already starts downregulation at E7.8 while H3K27me3 is not enriched yet at E7.5, suggesting that H3K27me3 itself does not play a role for the repression of Nodal. In line with this notion, I was wondering why H3K27me3 is maintained at high level around TSS in Usp9x mutant considering the finding that PRC2 could undergo strong poly-ubiquitination in the mutant. Authors should clarify why substantial amount of PRC2 can exist even in the Usp9x mutant and the mechanisms underlying thresholding of the level of PRC2. In my opinion, the distribution pattern of H3K27me3 is just the consequence rather than the cause of transcription. Authors should take into account those points and discuss in the text.

The Reviewer raises several excellent points of discussion. To the reviewer’s point about thresholding, we have added the following text to the Discussion (new text underlined): “Our data suggest that the decline in Usp9x expression at implantation contributes to destabilizing PRC2 to allow exit from pluripotency and lineage induction. Usp9x is not the only factor that sets the threshold for PRC2 activity at peri-implantation. Rather, we posit that post-translational modification augments other mechanisms that minimize H3K7me3 during this developmental

window, including decreased transcription of Suz12; rapid cell cycles that oppose H3K27me3 inheritance; and accumulation of H3K27me3 antagonists, including RNA, activating chromatin marks, and DNA methylation. Against this backdrop, PRC2 accessory proteins help recruit and retain the complex at CpG-rich regions such as bivalent promoters. As a result, even Usp9x-low ES cells retain H3K27me3 over peaks associated with promoters (Fig. 3c,d), reflecting concentrated PRC2 activity at these sites.”

We agree that H3K27me3 is one of several regulatory layers acting on developmental genes, and it may be that H3K27me3 accumulates in the wake of decreased transcription. We now specifically state this possibility in the Discussion: “De novo H3K27me3 may also accumulate in the wake of decreased transcription to reinforce repression.” Regarding H3K27me3 levels at the *Nodal* locus, we apologize to the Reviewer because in our original submission we inadvertently added a browser track from E7.5 instead of from E6.5 (Fig. 2g) – in other words, two replicates of E7.5 were shown. We have now corrected this, and it is clear that the mark is not detected at E6.5 but does appear over the locus above background by E7.5 (Fig. 2g), which coincides with its transcriptional downregulation (Fig. 2h), although there is a much larger increase in H3K27me3 between E7.5 and E8.5. We also note that these H3K27me3 ChIP-seq data are from wild-type embryos, not *Usp9x* mutants.

Minor points

1. Related to Major point 1. In line 52, although the authors attributed this re-distribution of H3K27me3 to switch in PRC2 activity, I think this could be overstated as grounds for this idea is not seems enough. Authors should modify texts.

Please see our response to major comment 2. We have updated the text to refer to the connection between *Usp9x* and global H3K27me3 level rather than a switch in the mode of complex activity.

2. I'm afraid enrich R is not yet common to all of us and thus Figure 1h right panel is not impressive. Rather, authors should show the distribution of PRC2 signals over promoters of DE (and other genes as negative controls) in the form of heatmaps.

We have updated Figure 1 to include heatmaps and profile plots demonstrating Suz12 enrichment over the 1310 genes upregulated in *Usp9x*-low ES cells compared to a random subset of the same size (new Fig. 1h). The graph with the EnrichR data has been moved to Supplementary Figure 2f.

3. I think the PCA analysis in Figure 2C is not so important. It'd be more convincing if authors showed the changes of expression of each 71 upregulated and 66 downregulated genes in the format of MA plot or so.

We thank the Reviewer for this suggestion. In the new Fig. 2c, we now show the top dysregulated genes across all mutants in the form of MA plots. These plots provide further evidence that the degree of differential gene expression in *Usp9x* mutants is relatively limited because we have captured the transcriptional landscape *before* the emergence of gross phenotype abnormalities, but several important developmental genes are dysregulated. In addition, we removed the original PCA plots and added a single PCA plot showing overall separation between mutants versus controls in both litters, shown in new Supplementary Fig. 3e, further supported by unsupervised hierarchical clustering in Supplementary Fig. 3f.

4, In the boxplots of Figure 2f right panel, the authors should state from where the signals are counted. Is that TSS±10 kb ?

The boxplot in Fig. 2f shows signal quantified over promoters, defined as TSS +10kb, -1kb. We have updated the figure legend to include this description and clarified in the methods section.

5. In Figure 3e, it is unclear what the x-axis signifies.

We have rearranged Fig. 3e (now Fig. 3f) and include an x-axis label under all 3 panels.

6. In Figure 3f, authors should explain how they calculated the p-value.

We have updated the legend for Figure 3 to clarify the statistics used to calculate p values in the figure.

7. In line 191 and 193, *Nodal* should be *Nodal*.

We have updated the text to italicize *Nodal* mRNA.

8. Authors should discuss what is the key player which promotes the ubiquitination of PRC2.

This is an excellent suggestion. Although it is difficult to say which of the published PRC2 ubiquitin ligases are critical during the window of peri-implantation development, the following underlined text has been added to the Discussion to speculate on this point:

“It will also be important to identify the E3 ubiquitin ligase(s) acting on PRC2. Several ligases have been found to ubiquitinate PRC2 in mammalian cells, including Praja1, CHIP/Stub1, Traf6, and Fbxw7. Our data reinforce that the balance between ubiquitin ligases and deubiquitinases dictates an important layer of PRC2 regulation during mouse development.”

Reviewer #3 (Remarks to the Author):

Here, Macrae and Ramalho-Santos examine the functions of Usp9x in ES cell pluripotency and embryonic development. Using an auxin-inducible degron allele of Usp9x, they compare the effects of Usp9x loss to controls and Usp9x-high cells that are somehow insensitive to auxin addition. They show that Usp9x is required for ES cell self-renewal and promotes a transcriptional state representative of naïve ES cells and pre-implantation embryos, while Usp9x-low cells are more similar to post-implantation embryos. More generally, the Usp9x-high cells exhibit global hypotranscription, and the genes most affected by Usp9x levels are enriched for PRC2 binding. Consistent with these findings, Usp9x KO embryos are developmentally delayed and appear to incompletely repress early lineage commitment genes. The authors address the molecular mechanisms underlying these phenotypes and show Usp9x-high ES cells have higher overall H3K27me3 levels genome-wide, similar to pre-implantation embryos. Loss of Usp9x leads to accumulation of ubiquitinated forms of PRC2 subunits, suggesting Usp9x increases PRC2 levels and H3K27me3 by deubiquitinating PRC2 subunits. Overall the manuscript is very thorough, using both ES cell and in vivo models to address the functions of Usp9x. The findings should be of significant interest to the stem cell and developmental biology communities. Although the experiments were largely well done and the data of high quality, I have a few questions and concerns, most notably regarding comparisons between Usp9x-high cells and other populations, as well as several minor points.

Major points:

1. Since many comparisons are made between control and Usp9x-high or -low cells, it is important to quantify Usp9x levels in each condition. However, the western blot in Fig. 1a, which suggests similar Usp9x levels between control and Usp9x-high cells, was run on two different gels. This prevents accurate comparison of the levels between control and Usp9x-high cells.

Additionally, quantification of relative Usp9x levels in high, low, and no-auxin samples would be helpful.

We apologize for inducing the Reviewer to misinterpret this figure panel: the samples shown in Fig. 1a were indeed all processed in parallel and run on the same gel; the vertical line denotes splicing out of an intervening lane from this gel. We have now specified this point in the figure legend and are able to quantify relative levels of Usp9x. We have added Supplementary Fig. 1b with this quantification.

2. Fig. 1g: If the Usp9x-high cells do indeed have levels of Usp9x that are similar to control cells, why do they exhibit hypotranscription relative to control cells? Unless I am missing something, this suggests the hypotranscription phenotype is independent of Usp9x levels. It is also strange that the hypotranscription of Usp9x-high cells relative to control cells is much stronger than the hypertranscription of Usp9x-low cells, which is very modest. Usp9x-low cells are strongly depleted for Usp9x, but have relatively modest changes in mRNA levels, while Usp9x-high cells have a strong reduction in global mRNA levels despite expressing similar Usp9x levels as control cells. How can this be explained?

Thank you for prompting us to improve our characterization of Usp9x-high ES cells. In the new Supplementary Fig. 2c we show MA plots to provide global comparisons of expression between Usp9x-high and Usp9x-low at all time points. Applying a standard cutoff ($\geq 1.5x$ change in expression and $\text{adj } p < 0.05$), we find ~ 4000 genes significantly downregulated in Usp9x-high and ~ 2500 genes significantly upregulated in Usp9x-low cells at 48h. Together with the new Supplementary Fig. 2d, we show that the hyper/hypotranscription phenotypes emerge after 48h without auxin, suggesting that additional transcriptional remodeling occurs after 8h auxin treatment and culture of the two populations in isolation. We agree that Usp9x level itself does not directly predict global transcriptional output and speculate that Usp9x marks distinct subpopulations of ES cells that diverge in their developmental potential, reflected in their transcriptional programs.

3. Similarly, in Fig. 3d, Usp9x-high cells exhibit higher H3K27me3 at repeat regions relative to no auxin controls. If Usp9x-high and no auxin controls have similar Usp9x levels, it is not clear what might be causing this increase in H3K27me3. Furthermore, it is odd that most of the comparisons in Fig. 3 are of Usp9x-high and Usp9x-low cells without the no-auxin control. Since the no-auxin control was profiled by ChIP-seq, it could be included in Fig. 3a-c, e-f. Because it is not clear why, on a molecular level, the Usp9x-high cells are resistant to auxin, comparisons with no auxin controls are important for the interpretation of these data.

We have updated Figure 3 to include the no-auxin population. As expected, the new plots show that the average population profile of H3K27me3 in no-auxin controls closely resembles the profile of Usp9x-low ES cells and is typical of serum ES cells. These results emphasize that the minority of Usp9x-high ES cells represent a distinct molecular state from Usp9x-low ES cells.

4. In Fig3b-c, the local background (H3K27me3 levels outside of the peak regions) is higher in Usp9x-high cells to the same degree that the aggregate peak enrichment is higher. Could this be due to a normalization artifact, or do regions at least 5 kb from peaks actually exhibit increased H3K27me3 in Usp9x-high cells at many locations?

We found that the global elevation in H3K27me3 signal occurs not only over canonical promoter-proximal peaks, but also spreading upstream and downstream of peaks as well as distal to peaks. If we further extend enrichment plots (as in Fig. 3c and 3d) beyond 5kb, we still see higher levels of H3K27me3 at the tails around peaks in Usp9x-high ES cells (“neg” = no auxin control):

This point is further supported by the global enrichment plots in Fig. 3f, right panel (which includes the entire genome divided into 10kb bins). While we cannot discount the possibility of normalization artifact, it is notable that our findings are closely aligned with recent studies documenting diffuse global accumulation of H3K27me3 beyond peaks in 2i ES cells (van Mierlo et al., *Cell Stem Cell*, 2019; Kumar et al., *Cell Reports*, 2019; Kurimoto et al., *Cell Stem Cell*, 2015). Visualizing coverage in the genome browser, sampled in Supplementary Fig. 4c, suggests that the enrichment is not uniform across the genome, although this local signal variation is obscured in our aggregate analyses in main Fig. 3. As noted in the end of this section of the Results: “The pattern of H3K27me3 in Usp9x-high serum ES cells resembles the diffuse domains of the mark in naïve (2i) ES cells and in pre-implantation embryos (Zheng et al., *Mol Cell*, 2016)”.

Minor points:

1. Fig. 1h: The number of genes observed and the number of genes expected from the 248 Usp9x affected that are targets of each regulator would be more informative than plots of their p-values, which have no biological meaning. Perhaps both sets of information could be included.

We thank the Reviewer for this suggestion. We updated Figure 1 to include plots of Suz12 ChIP-seq signal enrichment over Usp9x-associated genes to provide better representation of PRC2 enrichment over these genes (new Fig. 1h).

2. Fig. 2c: It is strange to show separate PCA plots for each litter. Presumably this is due to a batch effect that muddles the clustering, but plotting the data together would allow determination of exactly how well the two genotypes separate from each other in this analysis.

We initially included the separate PCA plots to remain consistent with our approach to identifying differentially expressed genes. Following up on the Reviewer’s suggestion, we removed the original PCA plots and added a single PCA plot showing overall separation between mutants versus controls in both litters, shown in new Supplementary Fig. 3e, further supported by unsupervised hierarchical clustering in Supplementary Fig. 3f. In addition, we now show the top dysregulated genes across all mutants in the form of MA plots. These plots provide further evidence that the degree of differential gene expression is relatively limited because we have captured the transcriptional landscape before the emergence of gross phenotypic abnormalities in mutants, but several important developmental genes are dysregulated.

3. Quantification of the H3K27me3 levels in each population in Fig. 3a would help for

understanding the magnitude of the global effect of Usp9x loss on this mark.

We have added a quantification to Fig. 3a. In addition to this western blot, the plot of ChIP recovery in (now Fig. 3b) provides further quantification of H3K27me3 levels in each cell population.

4. Fig. 4d: What is being immunoprecipitated and what antibody is being used in the western blot? The figure legend is not informative and the labels in the figure are lacking.

We thank the Reviewer for helping us improve figure clarity. We have updated the labeling in Fig. 4d to more clearly convey important experimental details.

5. For some panels, the y-axis labels are confusing. For example, “ $\log_2(\text{fold change}/\text{control})$ ” suggests the fold change value is further divided by the control value.

We have updated figure legends as suggested.

REVIEWER COMMENTS

Reviewer #1 (Remarks to the Author):

In this revised submission, the authors have attempted to address the major critiques raised in the initial round of review. A major contention of the manuscript, however, appears to be internally inconsistent.

The major and novel finding of the manuscript is that the USP9X deubiquitinase stabilizes PRC2 proteins in embryonic stem cells (ESCs) through its deubiquitinating activity. To test USP9X function, the authors deplete USP9X through an auxin-degron system. In the resultant ESCs, the authors find two distinct populations of ESCs: USP9X-high and USP9X-low. These two populations have differing transcriptomic signatures, with the USP9X-high ESCs displaying a more naïve ESC state and USP9X-low ESCs resembling more the primed pluripotent ESC state.

One important way to show USP9X dose regulating ESC state is through PRC2 regulation of specific loci, especially loci whose expression changes in USP9X-high vs. USP9X-low ESCs. The authors in fact provide such data comparing H3K27me3 occupancy in USP9X- low vs. USP9X-high vs. no-auxin treated control ESCs in Figure 3. However, there is a discrepancy between the H3K27me3 ChIP data in Figure 3 vs. the transcriptomic data in the same cells in Figure 1.

In Figure 3, H3K27me3 levels and occupancy across the genome are similar between the USP9X-low ESCs and no-auxin treated control ESCs and both are markedly different compared to USP9X-high ESCs. Whereas, In Figure 1a and 1f, the authors find that the transcriptome of USP9X-high ESCs is similar to the no-auxin treated control ESCs and both of these are now different than the USP9X-low ESCs. It's not clear how to reconcile these differing patterns of H3K27me3 deposition and gene expression states in the three classes of ESCs.

Other points:

1. Please emphasize better that the data in Figure 1h are not ChIP data on the Usp9x-low ESCs.
2. Although embryonic phenotypes aren't seen until embryonic day (E)9.5-E11.5 in mice lacking Usp9X, looking at transcriptional changes in earlier stage embryos would be a better follow-up to the ESCs. Especially since the model in Figure 5 makes suggestions about early post implantation stages E5.5 and E6.5.

Minor points:

1. Please specify in the legend what stage of embryo were sequenced in Figure 2c.
2. It is not clear what is the difference between the left and right side of Figure 2c. There seem to be two separate plots here based on the X axis.

Reviewer #2 (Remarks to the Author):

I agree that most of comments raised by all reviewers are adequately addressed and thus the revised manuscript is now suitable for publication in Nature Communications.

Reviewer #3 (Remarks to the Author):

The reviewers have satisfactorily addressed my previous concerns.

1. In Figure 3, H3K27me3 levels and occupancy across the genome are similar between the USP9X-low ESCs and no-auxin treated control ESCs and both are markedly different compared to USP9X-high ESCs. Whereas, In Figure 1a and 1f, the authors find that the transcriptome of USP9X-high ESCs is similar to the no-auxin treated control ESCs and both of these are now different than the USP9X-low ESCs. It's not clear how to reconcile these differing patterns of H3K27me3 deposition and gene expression states in the three classes of ESCs.

We thank the Reviewer for prompting us to clarify the relationship between Usp9x and PRC2. We note that it is reasonable to expect that linear relationships between Usp9x and PRC2 will be clearer in data pertaining to their direct biochemical interaction (i.e., Usp9x stabilizes and promotes protein levels of PRC2), whereas this relationship will likely be compounded by other factors in downstream effects, notably on H3K27me3 levels/distribution and the transcriptome. Despite this general consideration, there is indeed a remarkable correlation between Usp9x and PRC2 at all these levels, as we attempt to explain next.

We document that Usp9x is a PRC2 deubiquitinase that stabilizes complex members. Our biochemical reveal a dose-dependent relationship between Suz12/Ezh2 and Usp9x at the protein level (Fig. 4 and S5). The question of whether Usp9x level directly correlates with PRC2 activity may be confounded by other factors that influence histone methylation in the cellular context, e.g. the expression of antagonists such as histone demethylases, among other mechanisms. Even so, our data document that Usp9x expression correlates with PRC2 activity: H3K27me3 is more abundant in cells at the higher end of Usp9x expression (Usp9x-high) than at the lower end (Usp9x-low). No-auxin cells, which are effectively normal ES cells in serum/LIF, blend these two cell states but overall resemble the H3K27me3 profile of Usp9x-low ES cells. This suggests that the chromatin state of serum ES cells appears overall primed-like, which is in line with the expectation from the literature (Zheng et al., *Mol Cell*, 2016), with a minority of naïve-like cells carrying diffuse domains. In agreement with this notion, we wish to clarify that Usp9x-low ES cells do display a reduction in H3K27me3 relative to no-auxin cells. This can already be seen in the profile plots of H3K27me3 over peaks shown in both Fig. 3d and Supplementary Fig. 4d, although the effect is admittedly mild in this aggregate analysis. Prompted by the Reviewer's comment, we asked whether this reduction might be more obvious over genes. This analysis revealed a slight but consistent reduction of H3K27me3 over many genes, with 4-fold more regions showing $\geq 20\%$ reduction relative to no-auxin controls than reciprocal gain (Reviewer Fig. 1, left). Moreover, the Usp9x-low ES cells display a significant overall reduction in H3K27me3 levels at all genes relative to untreated cells (Reviewer Fig. 1, right). Therefore, in agreement with the expectations above regarding the biochemical relationship between Usp9x and PRC2 and heterogeneity of serum/LIF ES cells, Usp9x-hi ES cells display a major gain in H3K27me3 relative to no-auxin controls, whereas Usp9x-low ES cells display a relatively mild but significant reduction in H3K27me3 relative to no-auxin controls.

Reviewer Figure 1. Left: comparison of H3K27me3 reads counted over genes (+5kb upstream, counts log₂-transformed) in Usp9x-low or no-auxin ES cells. Dotted lines indicate ±20% change relative to no-auxin. Highlighted points are those that are >20% increased or decreased in both replicates, with *N* indicating the number of such genes. **Right:** boxplot quantification of H3K27me3 reads over genes (+5kb upstream, counts log₂-transformed) in the indicated cell states, showing overall highest level in Usp9x-high and mild reduction in Usp9x-low relative to no-auxin ES cells. P-value by Wilcoxon rank-sum test.

With regards to transcriptional output, it is of course subject to multiple layers of regulation, not just H3K27me3, and therefore would not necessarily be expected to have a clear one-to-one correlation with Usp9x level. Nevertheless, our data reveal that Usp9x-high cells represent a transcriptionally-repressed state, especially of PRC2 target genes, whereas Usp9x-low cells display widespread activation of PRC2 targets, in both cases relative to no-auxin controls. As shown in Figures 1, S1 and S2, several RNA-seq analyses establish that both Usp9x-high and Usp9x-low ES cells differ significantly from the no-auxin state:

(1) The GSEA plot shown in Fig. 1d is perhaps the best visual representation of how both Usp9x-high and Usp9x-low ES cells differ significantly from no-auxin controls. We realize that the fact that in both cases Usp9x-high or Usp9x-low ES cells were compared to no-auxin controls (not to each other) was not clear in the legend for Fig. 1d, and have now amended it to specifically state this. This plot shows significant induction of a pre-implantation-like transcriptional program in Usp9x-high cells and concomitant anti-correlation with post-implantation stages, relative to no-auxin controls; the converse is observed for Usp9x-low ES cells. qRT-PCR validates expression changes in Usp9x-high cells at representative genes (Supplementary Fig. 1f).

(2) Several plots support that Usp9x-high cells trend towards global suppression of steady-state transcript levels relative to no-auxin controls, starting at 8h but clearly visible at 48h of auxin treatment (Fig. 1i, Supplementary Fig. 2c,d,g). This is perhaps best visualized in MA plots of overall gene expression (Supplementary Fig. 2c) together with ribosomal protein gene expression (Supplementary Fig. 2d), which suggest a global effect belied by

the relatively limited number that meet our cutoff for differential expression (277 genes down, 71 genes up with fold-change > 1.5x, adjusted $P < 0.05$) – here again, relative to no-auxin controls.

To clarify this point, in the Results section we now note (new text underlined), “At 8h, Usp9x-high ES cells display mildly decreased expression of many transcripts relative to no-auxin control cells, although only 277 downregulated genes meet the threshold for differential expression (fold-change > 1.5 and adjusted $P < 0.05$), with 70 upregulated genes (Supplementary Fig. 2c). By 48h, Usp9x-high cells settle further into a state of hypotranscription, with suppression of the majority of the transcriptome relative to control cells (3941 significantly downregulated vs 38 upregulated, Fig. 1g and Supplementary Fig. 2c).”

(3) Global transcriptional differences are also evident in the PCA plot in Fig. 1c, where 8h Usp9x-high ES cells (in salmon) do cluster separately from no-auxin cells (in light gray), although we acknowledge that the labeling of the figure to encompass the 2 timepoints (8h and 48h) may have minimized this point.

In sum, Usp9x-high and Usp9x-low ES cells are both distinct from no-auxin controls, and in both cases the levels of Usp9x correlate positively with PRC2 protein levels, K27me3 levels and breadth of distribution, and repression of PRC2 target genes, when compared to no-auxin controls. We thank the Reviewer for their careful consideration of the impact of Usp9x on PRC2 protein levels, enzymatic activity, and downstream modulation of transcription. We hope that the Reviewer agrees that the discussion above and the revisions noted on the Results section and figure legends clarify this point.

Other/minor points:

2. Please emphasize better that the data in Figure 1h are not ChIP data on the Usp9x-low ESCs. We thank the reviewer for prompting us to clarify this point to the readers. We have made the suggested edit to Figure 1h legend: “Heatmaps with summary profile plot of Suz12 binding (mapped in wild-type ES cells⁵⁰) over the genes upregulated in Usp9x-low cells or a random subset ($N = 1310$).”

3. Although embryonic phenotypes aren’t seen until embryonic day (E)9.5-E11.5 in mice lacking Usp9X, looking at transcriptional changes in earlier stage embryos would be a better follow-up to the ESCs. Especially since the model in Figure 5 makes suggestions about early post implantation stages E5.5 and E6.5.

We agree that our data from ES cells point to the early post-implantation window (E5.5-E6.5) as key to the proposed Usp9x-PRC2 regulatory axis. The late time point (48h after auxin treatment) allows us to explore a window slightly beyond the peri-implantation window captured by serum ES cells (Supplementary Fig. 1,2). We therefore chose to focus our molecular analysis on the time period (E8.5) just before the emergence of morphological abnormalities (E9.5) in the Usp9x mutants. We have updated the legend for the model in Figure 5 to highlight that the data on premature lineage induction are from ES cells.

We agree that it will be of interest to explore the *in vivo* role of Usp9x in the immediate peri-implantation window. For the current study, we employed a Sox2-Cre in order to sidestep possible confounding from early pre-implantation or extra-embryonic roles of Usp9x, given suggestions from the literature that Usp9x plays roles in both contexts (Pantaleon et al., *Mech Dev*, 2002; Abed et al., *PLoS One*, 2019). This Cre is reported to be expressed in a minority of cells at E5.5 (Mulas et al., *Development*, 2018), and thus full Cre-mediated deletion and loss of the

corresponding protein is not expected to be complete by E6.5. Moreover, this approach allowed us to further explore the relationship between Usp9x and PRC2 at a stage where the role of PRC2 is much less well understood, that of silencing of lineage commitment genes post-gastrulation.

4. Please specify in the legend what stage of embryo were sequenced in Figure 2c.

We have updated the figure legend as suggested.

5. It is not clear what is the difference between the left and right side of Figure 2c. There seem to be two separate plots here based on the X axis.

The Reviewer is correct—these are separate plots, representing both litters (n = 3 mutant-control embryo pairs each) sequenced in the RNA-seq experiment. We apologize that this was not clear and have updated the figure and legend to reflect this.

REVIEWERS' COMMENTS

Reviewer #1 (Remarks to the Author):

I appreciate the authors response to my last round of critiques. I agree that USP9X is a novel regulator of PRC2 during embryonic development. This manuscript, therefore, adds another layer of how PRC2 function is orchestrated during development. The authors have done a credible job in responding to our comments.